# Formalizing Consistency and Coherence of Representation Learning

**Harald Strömfelt**
Department of Computing
Imperial College London
London, SW7 2AZ
h.stromfelt17@imperial.ac.uk

**Luke Dickens**
Department of Information Studies
University College London
London, WC1E 6BT
l.dickens@ucl.ac.uk

**Artur d'Avila Garcez**
Department of Computer Science
City, University of London
London, EC1V 0HB
a.garcez@city.ac.uk

**Alessandra Russo**
Department of Computing
Imperial College London
London, SW7 2AZ
a.russo@imperial.ac.uk

## Abstract

In the study of reasoning in neural networks, recent efforts have sought to improve consistency and coherence of sequence models, leading to important developments in the area of neuro-symbolic AI. In symbolic AI, the concepts of consistency and coherence can be defined and verified formally, but for neural networks these definitions are lacking. The provision of such formal definitions is crucial to offer a common basis for the quantitative evaluation and systematic comparison of connectionist, neuro-symbolic and transfer learning approaches. In this paper, we introduce formal definitions of consistency and coherence for neural systems. To illustrate the usefulness of our definitions, we propose a new dynamic relation-decoder model built around the principles of consistency and coherence. We compare our results with several existing relation-decoders using a partial transfer learning task based on a novel data set introduced in this paper. Our experiments show that relation-decoders that maintain consistency over unobserved regions of representation space retain coherence across domains, whilst achieving better transfer learning performance.

## 1 Introduction

Humans are capable of learning concepts that can be applied to many different scenarios [18, 34, 23]. An important principle is that human-like concepts remain *coherent* across contexts [31]. As an example, consider the concept of *ordinality*, *e.g.* "A is larger than B", which allows comparisons to be made between ordered sets. Ordinality should apply equally whether A and B are digits or a tower of blocks. It is said that a concept may pertain to a multitude of properties: position, volume, reach, *etc*. As long as one of these properties can be attributed to an object, a set of objects can be compared on that basis. All in all, if the concept of ordinality was to be learned in its most general form, its use should be consistent across objects and coherent across object properties.

In [31], empirical results on story generation and instruction-following have shown that an intuitive use of consistency and coherence can increase the accuracy of neural networks. Following a neuro-symbolic perspective [12], it is argued in [31] that *System 1* approaches, fast and capable of learning

---

*Funding in direct support of this work: EPSRC Training Grant, project reference EP/L504786/1.

36th Conference on Neural Information Processing Systems (NeurIPS 2022).

patterns efficiently from data, "are often inconsistent and incoherent", and that "adding *System 2*-inspired logical reasoning" as a logical consistency, training-free module allows for an improved selection of candidate stories generated by *System 1*. While [31] makes an important contribution by exploring several variations on the theme, in this paper we offer a formal definition for consistency and coherence in the context of neural networks, in particular neuro-symbolic autoencoders. We also apply and evaluate consistency and coherence in transfer learning tasks, where we believe that the theme will have its most practical impact.

We argue that for a concept to be useful during transfer learning, the system of relations that define the concept in the source domain must be coherent with the target domain, whereby logical consistency achieved in the source is retained in the target domain. This is to say that the concept-specific relations learned in the source ought to be consistent with a logical theory that defines their semantics, and that such consistency must extend beyond the representations learned in the source domain and, in particular, hold for the embeddings learned in the target domain.

In this paper, we offer a formal definition for consistency and coherence of sub-symbolic representation learners, inspired by analogous definitions from symbolic AI. This is expected to define the conditions that make a learned concept transfer well across properties and objects. To evaluate the practical value of these definitions in a real setting, we derive a simple neuro-symbolic autoencoder architecture consisting of a neural encoder for objects coupled with consistent modular object relation-decoders. Relations such as isGreater, isEqual,... are evaluated on a proposed Partial Relation Transfer (PRT) learning task, between a new CLEVR-style BlockStacks data set and the MNIST handwritten digits data set, such that the learning of ordinality among the MNIST digits is evaluated against the learning of the relative position of a red block in a stack of multi-colored blocks. Our evaluation includes a comparison with several existing relation-decoder models and results show that relation-decoders which maintain consistency over unobserved regions of representation space retain coherence across domains whilst achieving better transfer learning performance.[1] In summary, the contributions of this paper are:

- A formal definition of consistency and coherence for sub-symbolic learners offering a practical evaluation score for concept coherence;

- A derived model implementation and PRT data set and experimental setup used to evaluate the interplay between concept coherence and concept transfer;

- A comprehensive critical evaluation of results and comparison of multiple relation-decoder models, showing that improvements in concept coherence, as defined in this paper, correspond with improved concept transfer.

In Section 2 we provide the notation and required logic background. Section 3 formally defines coherence and consistency. Section 4 defines a practical consistency loss and Section 5 outlines our neuro-symbolic autoencoder. After detailing the PRT task and introducing the data set in Section 6, comparative experimental results are discussed in Section 7. We provide an overview of the related work in Section 8 and Section 9 concludes the paper with a discussion, including limitations and future work. We expand on the experimental results and setup, together with data set characteristics, model details and parameterization in the Appendices.[2]

## 2 Preliminaries

**Notation:** We reserve uppercase calligraphic letters to denote sets, and lowercase versions of the same letter to denote their elements, e.g. $\mathcal{S} = \{s_1, \ldots, s_n\}$ is a set $\mathcal{S}$ of $n$ elements $s_i$. We indicate with $|\mathcal{S}| = n$ the cardinality of $\mathcal{S}$. We use uppercase roman letters to denote a random variable (e.g. S), and use the uppercase calligraphic version of the same letter ($\mathcal{S}$) to denote the set from which the random variable takes values according to some corresponding probability distribution $p_S$, over the elements of the set, such that $\sum_{i=1}^{|\mathcal{S}|} p_S(s_i) = 1$ for a discrete $\mathcal{S}$. For brevity, we may write $p_S(s_i)$ as $p(s_i)$, where the random variable is implied by the argument. We use bold font lowercase letters to denote vector elements, e.g. $\boldsymbol{s}_i \in \mathbb{R}^d$ is an d-dimensional vector element from the set $\mathcal{S} = \mathbb{R}^d$.

---

[1]This paper formalizes the theory and extends the empirical results first reported in [43].

[2]The codebase for this paper can be found at `https://github.com/HStromfelt/neurips22-FCA`.

**Logic and model-theoretic background:** our proposed theory is based upon logic and model theoretic primitives. To avoid making this paper overly dense, we defer the details of the logic background to Appendix E and include here only the most important definitions supported by an illustrative example.

**Definition 2.1** (Signature, Arity, Domain, Interpretation, Structure). The *signature* of a language $\mathcal{L}$ is a set of relations $\sigma = \{r \in \mathcal{L}\}$ whose elements have *arity* given by $\mathsf{ar} : \sigma \to \mathcal{N}$, where $\mathcal{N}$ is the set of natural numbers. Given a signature $\sigma$ and a non-empty *domain* $\mathcal{S} = \{s_1, s_2, \ldots\}$, an *interpretation* $I_{\mathcal{S}_\sigma}$ of $\sigma$ over elements of $\mathcal{S}$ assigns to each relation $r \in \sigma$ a set $I_{\mathcal{S}_\sigma}(r) \subseteq \mathcal{S}^{\mathsf{ar}(r)}$. A *structure* is a tuple $\mathcal{S}_\sigma = (\mathcal{S}, I_{\mathcal{S}_\sigma})$.

We construct universally quantified first-order logic formulae (called sentences) using the signature of $\mathcal{L}$. A set of sentences form a theory $\mathcal{T}$ and when a sentence $\tau$ is true in a structure $\mathcal{S}_\sigma$, we say that the structure satisfies $\tau$, denoted as $\mathcal{S}_\sigma \models \tau$. This allows us to define a *model* of a theory:

**Definition 2.2** (Model of a theory). Let $\mathcal{T}$ be a theory written in a language $\mathcal{L}$ and let $\mathcal{S}_\sigma = (\mathcal{S}, I_{\mathcal{S}_\sigma})$ be a structure, where $\sigma$ is the signature of $\mathcal{L}$. $\mathcal{S}_\sigma$ is a *model of* $\mathcal{T}$ if and only if $\mathcal{S}_\sigma \models \tau$ for every sentence $\tau \in \mathcal{T}$.

*Example* 1. Let $\mathcal{S}$ is a domain of images of handwritten digits and $\sigma$ the signature of binary relations $\sigma = \{\mathsf{isGreater}, \mathsf{isEqual}, \mathsf{isLess}, \mathsf{isSuccessor}, \mathsf{isPredecessor}\}$, or for short $\sigma = \{\mathsf{G}, \mathsf{E}, \mathsf{L}, \mathsf{S}, \mathsf{P}\}$. Let $\mathcal{T}$ be the theory that defines ordinality including, for instance, the sentence $\forall i, j. \, \mathsf{G}(i, j) \to \neg \mathsf{E}(i, j)$ (if a digit is greater than another then they are not equal). Any structure $\mathcal{S}_\sigma = (\mathcal{S}, I_{\mathcal{S}_\sigma})$ with interpretations $I_{\mathcal{S}_\sigma}$ of $\sigma$ that captures a total order over the elements of $\mathcal{S}$ is a model of $\mathcal{T}$.

# 3 A Formalization of Consistency and Coherence

In this section, we turn our attention to the challenge of learning a model of a theory (Def. 2.2) over a real-world domain $\mathcal{S}$ given a signature $\sigma$. Here, a learner must determine an appropriate interpretation over real-world data, such as images or other perceptions. This can be challenging because, firstly, we may only have a partial description of the interpretation, and secondly data may be noisy and contain information that is not relevant to the theory. For example, the handwritten digits in the MNIST data set contain stylistic details such as line thickness and digit skew that are irrelevant to the notion of ordinality, which makes learning the structure from Example 1 non-trivial.

Following the convention from the autoencoder disentanglement literature [4, 21, 17, 16], we make the assumption that real-world observations S are drawn from some conditional distribution $p_{\mathsf{S}|\mathsf{Z}}$, where Z is a latent random variable, itself drawn from prior $p_{\mathsf{Z}}$. It is therefore useful to define a domain *encoding* of the form:

$$\psi_{\mathcal{S}} : \mathcal{S} \to \mathcal{Z}, \tag{1}$$

tasked with approximating the conditional expectation of the posterior, *i.e.* $\psi_{\mathcal{S}}(s) = \mathbb{E}_{p_{\mathsf{Z}|S}}[\mathsf{Z}|s]$. Since obtaining an interpretation from domain encodings for a given signature may require dealing with noise, we express the interpretation of relations over real-world data by belief functions [33, 32] over the space $\mathcal{Z}$, and refer to these as *relation-decoders*:

$$\phi_r : \mathcal{Z}^{\mathsf{ar}(r)} \to (0, 1) \tag{2}$$

with $\phi = \{\phi_r : r \in \sigma\}$. Concretely, for a binary relation $r$ and ordered pair $(s_i, s_j) \in \mathcal{S}^2$, $\phi_r(\psi_{\mathcal{S}}(s_i), \psi_{\mathcal{S}}(s_j))$ describes the belief that $(s_i, s_j) \in I_{\mathcal{S}_\sigma}(r)$. A belief $\phi_r(\psi_{\mathcal{S}}(s_i), \psi_{\mathcal{S}}(s_j)) \approx 1$ signifies a strong belief that $(s_i, s_j) \in I_{\mathcal{S}_\sigma}(r)$ and $\phi_r(\psi_{\mathcal{S}}(s_i), \psi_{\mathcal{S}}(s_j)) \approx 0$ signifies a strong belief that $(s_i, s_j) \notin I_{\mathcal{S}_\sigma}(r)$. Together, $\psi_{\mathcal{S}}$ and $\phi$ allow us to define a belief-based analogue to a structure.

**Definition 3.1** (Soft-Structure/Soft-Substructure). Given a signature $\sigma$, a (possibly infinite) set $\mathcal{Z}$ and relation-decoders $\phi$, a *soft-structure* is a tuple $\tilde{\mathcal{Z}}_\sigma = (\mathcal{Z}, \phi)$. For a finite domain $\mathcal{S}$ and encoding $\psi_{\mathcal{S}} : \mathcal{S} \to \mathcal{Z}$, $\tilde{\mathcal{S}}_\sigma = (\psi_{\mathcal{S}}(\mathcal{S}), \phi)$ is called a finite *soft-substructure* of $\tilde{\mathcal{Z}}_\sigma$, with sub-domain $\psi_{\mathcal{S}}(\mathcal{S}) = \{\psi_{\mathcal{S}}(s) | s \in \mathcal{S}\} \subseteq \mathcal{Z}$.

A soft-structure can be used to learn a logic structure over a real-world domain through learning $\psi_{\mathcal{S}}$ and $\phi$. Clearly, a finite soft-substructure is a soft-structure. In a real-world domain, there may be only partial information about the values of an interpretation, and there may be errors in that partial

interpretation. To determine the degree to which a soft-structure *supports* any given structure, we introduce the following measure:

$$p(\mathcal{S}_\sigma | \tilde{\mathcal{S}}_\sigma) = \prod_{r \in \sigma} \prod_{O \in \mathcal{S}^{\mathrm{ar}(r)}} f(\phi_r, \psi_\mathcal{S}, O, \gamma^r_{O,\mathcal{S}_\sigma}) \tag{3}$$

with

$$f(\phi_r, \psi_\mathcal{S}, O, \gamma^r_{O,\mathcal{S}_\sigma}) = (\phi_r(\psi_\mathcal{S}(O)))^{\gamma^r_{O,\mathcal{S}_\sigma}} \cdot (1 - \phi_r(\psi_\mathcal{S}(O)))^{1 - \gamma^r_{O,\mathcal{S}_\sigma}}, \tag{4}$$

where $\gamma^r_{O,\mathcal{S}_\sigma} = 1$ if $O \in I_{\mathcal{S}_\sigma}(r)$, and $0$ otherwise; we use $\phi_r(\psi_\mathcal{S}(O))$ as shorthand for $\phi_r(\psi_\mathcal{S}(s_1), \ldots, \psi_\mathcal{S}(s_n))$ for $n = \mathrm{ar}(r)$. Eqn. 3 expresses the assumption that, given a finite soft-structure, the beliefs in what constitutes the interpretations of different relations are independent of one another. It is straightforward to show that $\sum_{\mathcal{S}_\sigma} p(\mathcal{S}_\sigma | \tilde{\mathcal{S}}_\sigma) = 1$ (summed over all possible structures with domain $\mathcal{S}$ and signature $\sigma$) and so Eqn. 3 can be treated as a probability measure, where $p(\mathcal{S}_\sigma | \tilde{\mathcal{S}}_\sigma) \approx 1$ means that there is a high probability that the interpretation sampled from $\tilde{\mathcal{S}}_\sigma$ will be $I_{\mathcal{S}_\sigma}$. If we have a theory $\mathcal{T}$ over $\sigma$ then it is natural to ask with what weight $\tilde{\mathcal{S}}_\sigma$ supports any given structure that is a model of $\mathcal{T}$. In the following, we use *model weight*, $\Gamma_\mathcal{T}^{\tilde{\mathcal{S}}_\sigma}$, to describe the support given by $\tilde{\mathcal{S}}_\sigma$ to models of $\mathcal{T}$:

$$\Gamma_\mathcal{T}^{\tilde{\mathcal{S}}_\sigma} = \sum_{\mathcal{S}_\sigma \in \mathcal{M}_\mathcal{S}^\mathcal{T}} p(\mathcal{S}_\sigma | \tilde{\mathcal{S}}_\sigma) \tag{5}$$

where $\mathcal{M}_\mathcal{S}^\mathcal{T}$ is the set of all structures with domain $\mathcal{S}$ that are models of $\mathcal{T}$. This lets us compare soft-structures, wherein a good soft-structure will be one that has a high model weight.

**Definition 3.2** ($\epsilon$-Consistency of soft-structures)**.** Given a finite soft-structure $\tilde{\mathcal{S}}_\sigma$ and an arbitrarily small number $\epsilon$, if $1 - \Gamma_\mathcal{T}^{\tilde{\mathcal{S}}_\sigma} \leq \epsilon$ then we say that $\tilde{\mathcal{S}}_\sigma$ is $\epsilon$-*consistent* with theory $\mathcal{T}$.

We propose $\epsilon$-consistency as an appropriate measure of the notion of consistency presented in [31]. A consistent soft-structure $\tilde{\mathcal{S}}_\sigma$ ensures that $\phi$ gives high belief only to interpretations that satisfy, and therefore are logically consistent with, theory $\mathcal{T}$. As expected, consistency pertains to the domain encodings of $\tilde{\mathcal{S}}_\sigma$, *i.e.* $\psi_\mathcal{S}(\mathcal{S})$. For a concept to be learned in a manner comparable to how a human might learn, we would expect consistency to carry over to new domains with their corresponding soft-structures, which motivates our definition of coherence between soft-structures, as follows. Consider a situation where a deep network has already learned from the MNIST data set a soft-structure that has high model weight, given the relations $\{\mathsf{G}, \mathsf{E}, \mathsf{L}, \mathsf{S}, \mathsf{P}\}$ from Example 1. Now, consider a new domain of images, $\mathcal{Y}$, showing single block stacks of different heights, and we wish to re-use the signature of ordinal relations and $\mathcal{T}$ from Example 1. Let $I_{\mathcal{Y}_\sigma}$ be an interpretation in the new domain that orders images according to block stack height and that is a model of $\mathcal{T}$. We can summarise this with the following two structures:

$$\mathcal{X}_\sigma = (\mathcal{X}, I_{\mathcal{X}_\sigma}) \in \mathcal{M}_\mathcal{X}^\mathcal{T} \quad \text{and} \quad \mathcal{Y}_\sigma = (\mathcal{Y}, I_{\mathcal{Y}_\sigma}) \in \mathcal{M}_\mathcal{Y}^\mathcal{T}, \tag{6}$$

where $\mathcal{X}_\sigma$ is the structure from Example 1 with a domain of handwritten digits and $\mathcal{Y}_\sigma$ is our new structure, with a domain of block stack images. These can be learned by soft-structures:

$$\tilde{\mathcal{X}}_\sigma = (\psi_\mathcal{X}(\mathcal{X}), \phi) \quad \text{and} \quad \tilde{\mathcal{Y}}_\sigma = (\psi_\mathcal{Y}(\mathcal{Y}), \phi), \tag{7}$$

which use domain-specific encoders, $\psi_\mathcal{X}$ and $\psi_\mathcal{Y}$, but share the same relation-decoders. As we know that $\tilde{\mathcal{X}}_\sigma$ has a high model weight and since $\phi$ is shared with $\tilde{\mathcal{Y}}_\sigma$, a natural question to ask is: under what conditions will a $\phi$ that is consistent over domain-encodings $\psi_\mathcal{X}(\mathcal{X})$ also be consistent over $\psi_\mathcal{Y}(\mathcal{Y})$? Concretely, we are interested in specifying when the following *coherence* condition holds.

**Definition 3.3** ($\epsilon$-Coherence across soft-structures)**.** Two soft-structures, $\tilde{\mathcal{X}}_\sigma$ and $\tilde{\mathcal{Y}}_\sigma$ that share relation-decoders $\phi$, are said to be $\epsilon$-*coherent* with respect to a theory $\mathcal{T}$, if $\tilde{\mathcal{X}}_\sigma$ is $\epsilon_1$-consistent with $\mathcal{T}$, $\tilde{\mathcal{Y}}_\sigma$ is $\epsilon_2$-consistent with $\mathcal{T}$, $\epsilon_1 \leq \epsilon$, and $\epsilon_2 \leq \epsilon$.

Coherence between $\tilde{\mathcal{X}}_\sigma$ and $\tilde{\mathcal{Y}}_\sigma$ as defined above means that the concept of ordinality that applies to digit ordering can also be applied to block stack height ordering. It is desirable that learning ordinality on the domain of digits produces a coherent concept of ordinality with respect to other ordinal properties, such as height. Since it is possible that $\psi_\mathcal{S}(\mathcal{X})$ and $\psi_\mathcal{S}(\mathcal{Y})$ produce unique encodings, coherence relies on $\phi$'s ability to generalize over possibly disjoint subsets of $\mathcal{Z}$.[3]

---

[3]If soft-structure $\tilde{\mathcal{Z}}_\sigma$ defined over the full space $\mathcal{Z}$ is consistent then coherence is guaranteed between all possible soft-substructures.

# 4 Measuring Consistency and Coherence

Calculating Eqn. 5 can be computationally too expensive for larger domains. An efficient approach to measuring the consistency of soft-structures is therefore required. In this section, we introduce a proxy measure for a soft-structure's $\epsilon$-consistency and $\epsilon$-coherence with a given theory when access to every logical model is not available or is computationally intractable.

Suppose that there is a fixed domain $\mathcal{S}$ and theory $\mathcal{T}$ whose sentences use relations from a signature $\sigma$. Let $k \in \{1, ..., K_0\}$ denote the index associated with each unique ground instance of the sentences in $\mathcal{T}$. Take $B_{\mathcal{T}}$ to be a Boolean random variable. For the $k$-th grounding in $\mathcal{T}$, the probability of the theory being satisfied under soft-structure $\tilde{\mathcal{S}}_\sigma$ is expressed as $p(b_{\mathcal{T}} = 1|\tilde{\mathcal{S}}_\sigma, k)$. Conversely, the probability of non-satisfaction is given by $p(b_{\mathcal{T}} = 0|\tilde{\mathcal{S}}_\sigma, k)$. For a model of (universally-quantified) theory $\mathcal{T}$, $\mathcal{S}_\sigma \in \mathcal{M}_{\mathcal{S}}^{\mathcal{T}}$, any grounding $k$ of the $\mathcal{T}$ is always satisfied (by definition), and thus $p(b_{\mathcal{T}} = 1|\mathcal{S}_\sigma, k) = 1$. When $\tilde{\mathcal{S}}_\sigma$ is consistent with $\mathcal{T}$ then we should also find that $p(b_{\mathcal{T}} = 1|\tilde{\mathcal{S}}_\sigma, k) \approx 1$ for all $k$. Hence, we define a consistency loss function as the expectation over a randomly-chosen grounding $k$ of the binary cross-entropy between $p(B_{\mathcal{T}}|\mathcal{S}_\sigma, k)$ and $p(B_{\mathcal{T}}|\tilde{\mathcal{S}}_\sigma, k)$ for any $\mathcal{S}_\sigma \in \mathcal{M}_{\mathcal{S}}^{\mathcal{T}}$. This in turn simplifies to produce the expected negative log-likelihood of satisfying a random grounding of $\mathcal{T}$, as follows:

$$L(\mathcal{T}, \tilde{\mathcal{S}}_\sigma) = \mathbb{E}_{k \sim p(k)}[-\ln p(b_{\mathcal{T}} = 1|\tilde{\mathcal{S}}_\sigma, k)]. \tag{8}$$

where $p(k) = \frac{1}{K_0}$ is the uniform distribution over the set of unique groundings. A measure based on this loss is required to enable a practical evaluation of consistency, acting as an approximation for consistency. More precisely, we define $\bar{\Gamma}_{\mathcal{T}}^{\tilde{\mathcal{S}}_\sigma} = \exp(-L(\mathcal{T}, \tilde{\mathcal{S}}_\sigma))$ as a proxy measure of $\Gamma_{\mathcal{T}}^{\tilde{\mathcal{S}}_\sigma}$, and say that soft-structure $\tilde{\mathcal{S}}_\sigma$ is $\bar{\epsilon}$-proxy consistent with $\mathcal{T}$ if

$$\ln \frac{1}{1 - \bar{\epsilon}} \geq L(\mathcal{T}, \tilde{\mathcal{S}}_\sigma) \tag{9}$$

where $\bar{\epsilon} \geq 1 - \bar{\Gamma}_{\mathcal{T}}^{\tilde{\mathcal{S}}_\sigma}$. Due to the relationship between $\bar{\epsilon}$ and $L(\mathcal{T}, \tilde{\mathcal{S}}_\sigma)$, we take the proxy measure of coherence to be the smallest satisfiable value of $L(\mathcal{T}, \tilde{\mathcal{S}}_\sigma)$ between domains.[4]

Although our treatment of consistency has thus far focused on a particular theory $\mathcal{T}$, notice that a subset of the sentences of $\mathcal{T}$ form a partial theory, which is itself a theory. This means that consistency can be evaluated given a partial (even single sentence) theory, allowing us to examine consistency losses for any partial specifications of a given domain of interest. In this paper, we evaluate the proposed consistency loss against two partial specifications within the theory of ordinality. These are named Consistency-Across (Con-A) and Consistency-Individual (Con-I) in the experiments that will follow.[5] Con-A includes the sentences that determine inter-relation behavior, for instance the sentence $\forall i, j.\ (\mathsf{isGreater}(i, j) \rightarrow \neg\mathsf{isEqual}(i, j) \wedge \neg\mathsf{isLess}(i, j))$, stating that if $i$ is greater than $j$ then $i$ must not be equal to or less than $j$. Con-I includes the sentences that are about a single relation, describing any property of an individual relation over objects (or images in the case of our experiments). Each relation may satisfy a number of properties, for example $\forall i, j, k.\ (\mathsf{isGreater}(i, j) \wedge \mathsf{isGreater}(j, k) \rightarrow \mathsf{isGreater}(i, k))$ represents transitivity of the $\mathsf{isGreater}$ relation. Transitivity is true for $\mathsf{isGreater}$, but is false for other relations investigated in this paper, e.g. $\mathsf{isSuccessor}$. We will evaluate consistency loss of transitivity (Con-I-T), asymmetry (Con-I-A) and reflexivity (Con-I-R) for the relations in Example 1. The evaluation of consistency loss for any available partial theory will be shown to provide a more nuanced perspective on model performance than accuracy results and disentanglement pressure alone during transfer learning.

# 5 A Consistent and Coherent Neuro-symbolic Autoencoder

In order to ground our definitions of consistency (Def. 3.2) and coherence (Def. 3.3) into a real system and evaluate their practical value, in this section we derive a simple neuro-symbolic autoencoder architecture which offers one of many possible implementations of the theory defined in Section 3. Figure 1 outlines the main components of our autoencoder: a domain-encoder $\psi_{\mathcal{S}}$ and modular

---

[4]The complete derivation of loss function and bounds is presented in Appendix G.

[5]Truth-tables for each consistency formula are given in Appendix F.

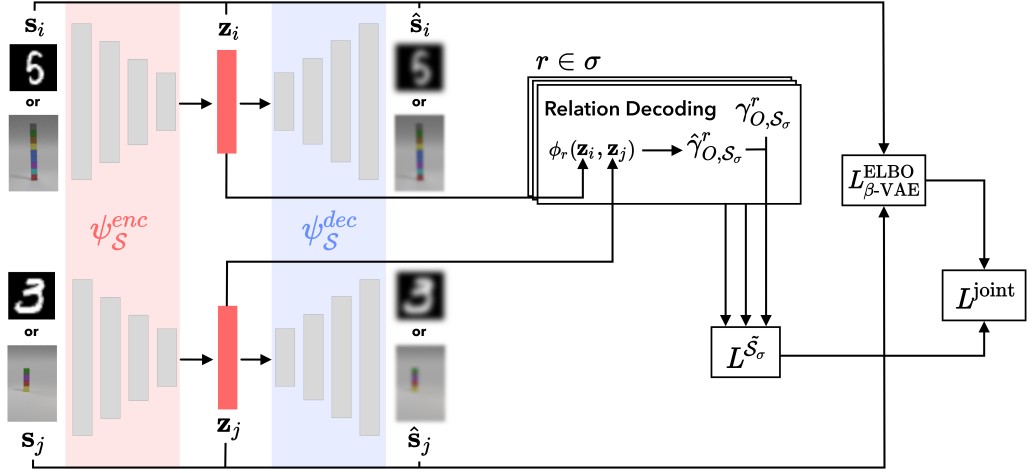

Figure 1: Network architecture used for PRT tasks. In our experiments $s_{i/j}$ are either MNIST (domain $\mathcal{X}$) or BlockStacks (domain $\mathcal{Y}$) images. Relational learning is performed on the source ($\mathcal{S} = \mathcal{X}$) MNIST domain (to learn e.g. that digit 5 is greater than 3). Moving to the target ($\mathcal{S} = \mathcal{Y}$) domain (stacks of blocks) involves training a new image autoencoder together with a subset of the relation-decoders from MNIST with fixed parameters. The remaining relations are held-out to evaluate zero-shot transfer learning performance. $\gamma^r_{O,\mathcal{S}_\sigma}$ provides the ground truth for the given structure, which $\phi_r$ predicts as $\hat{\gamma}^r_{O,\mathcal{S}_\sigma}$. As in Section 3, $O$ is used to abbreviate $(\psi^{enc}_{\mathcal{S}}(s_i), \psi^{enc}_{\mathcal{S}}(s_j))$.

relation-decoders $\phi$ form an autoencoding architecture that, given a domain of images $\mathcal{S} \subset \mathbb{R}^{C \times W \times H}$ ($C$ color channels, width $W$ and height $H$) and a $d$-dimensional latent space $\mathcal{Z} = \mathbb{R}^d$, converts sub-symbolic encodings from $\psi_{\mathcal{S}}$ into a modular relational representation via decoding for each $\phi_r, r \in \sigma$. Additionally, to retain information in $\mathcal{Z}$ pertaining to $\mathcal{S}$ which is beyond the requirements of $\phi$, a domain-decoder produces domain reconstructions $\hat{\mathcal{S}}$. In Figure 1, we use $\psi^{enc}_S$ to refer to the domain-encoder and $\psi^{dec}_S$ to the domain-decoder. Although in this paper we opt for an autoencoding architecture, our definitions of consistency and coherence are applicable to a wider range of neural architectures. For instance, a multi-layer perception network can be viewed as a set of encoding and decoding layers [40]. As long as the architecture offers explicit soft relation decodings, provided we can define a partial theory over them, we can define a consistency loss over the outputs.

To train the model, ground-truth interpretations $I_{\mathcal{S}_\sigma}$ are provided, allowing us to maximize directly Eqn. 3 via the negative log-likelihood loss:

$$L^{\tilde{\mathcal{S}}_\sigma} = -\log p(\mathcal{S}_\sigma | \tilde{\mathcal{S}}_\sigma), \tag{10}$$

To obtain informative latent representations for $\mathcal{S}$, we use a Variational Autoencoder (VAE), specifically the $\beta$-VAE [6, 17, 21] due to its simplicity and demonstrated ability to separate distinct factors in the latent representation (known as *disentanglement*, although disentanglement is not a requirement for consistency and coherence). We therefore take the Evidence Lower Bound (ELBO) objective with an additional $\beta$ scalar hyperparameter from [17], that seeks to achieve disentanglement ($L^{\text{ELBO}}_{\beta\text{-VAE}}$), and combine it with $L^{\tilde{\mathcal{S}}_\sigma}$ to obtain the following aggregate objective:[6]

$$L^{\text{joint}} = L^{\text{ELBO}}_{\beta\text{-VAE}} - \lambda L^{\tilde{\mathcal{S}}_\sigma} \tag{11}$$

where $\lambda$ is a scalar weighting parameter.

Together with the $L^{\text{ELBO}}_{\beta\text{-VAE}}$, the choice of relation-decoder can shape the domain encodings [15]. In our evaluation, the following choices are made. We propose a Dynamic Comparator (DC) composed of two modes, a distance-based measure, $\phi^\dagger_r$, to measure the distance between two inputs relative to a reference point, and a step-function, $\phi^\ddagger_r$, that determines the sign of the difference between two points, optionally with an offset. Although any function could be used that has the required characteristics for $\phi^\dagger$ and $\phi^\ddagger$, in this paper we use the following implementation:

$$\phi^{DC}_r(z_i, z_j) = a_{r,0} \cdot \phi^\dagger_r + a_{r,1} \cdot \phi^\ddagger_r \tag{12}$$

---

[6]a more detailed derivation of $L^{\text{ELBO}}_{\beta\text{-VAE}}$ is included in the Appendix C

where,

$$\phi_r^\dagger = f_0\left(-\eta_{r,0} \cdot \|\boldsymbol{u}_r \odot (\boldsymbol{z}_i - \boldsymbol{z}_j + \boldsymbol{b}_r^\dagger)\|_2\right) \tag{13}$$

$$\phi_r^\ddagger = f_1\left(\eta_{r,1} \cdot \boldsymbol{u}_r^\top (\boldsymbol{z}_i - \boldsymbol{z}_j + \boldsymbol{b}_r^\ddagger)\right). \tag{14}$$

Here, $\boldsymbol{a}_r = \texttt{Softmax}(\boldsymbol{A}_r) \in (0,1)^2$ is an attention weighting between the two modes, $\phi_r^\dagger$ and $\phi_r^\ddagger$; $f_0$ and $f_1$ are an exponential and sigmoid function, respectively; $\boldsymbol{u}_r = \texttt{Softmax}(\boldsymbol{U}_r) \in (0,1)^m$ is an attention mask which is applied to $m$-dimensional embeddings; $\boldsymbol{b}_r^\dagger, \boldsymbol{b}_r^\ddagger \in \mathbb{R}^m$ are learnable bias terms that enable an offset to each mode; $\eta_{r,0} \in \mathbb{R}^+$ are non-negative and $\eta_{r,1} \in \mathbb{R}$ are any-valued scalar terms, respectively. Lastly, $\odot$ denotes the Hadamard product and $\|\cdot\|_2$ is the Euclidean norm. The key innovation behind DC is its ability to model each of the ordinality relations whilst encouraging generalized consistency across the full latent subspace, as defined by each $\boldsymbol{u}_r$. This is achieved without explicit weight sharing, wherein relation-decoders discover parametric relationships from the data. Further details are provided in Appendix D.1.

## 6 Experiment Design: Partial Relation Transfer

We now describe an experimental design to compare coherence of different relation-decoders.

**Partial Relation Transfer (PRT):** We evaluate a novel PRT task across two soft-structures $\tilde{\mathcal{X}}_\sigma$ and $\tilde{\mathcal{Y}}_\sigma$. The soft-structures share a common signature $\sigma$ and relation-decoders $\phi$, but have disjoint domains $\mathcal{X}$ and $\mathcal{Y}$, respectively. The experimental design involves first learning $\phi$ on source domain $\mathcal{X}$, together with its domain-specific autoencoder. Then, a new domain-specific autoencoder is trained on the target domain $\mathcal{Y}$, alongside a selection of the now learned $\phi$ relation-decoders with fixed-parameters. The selection of relation-decoders is expected to help guide training of $\psi_{\mathcal{Y}}^{\text{enc}}$ (see Fig.1). Held-out relation-decoders are then evaluated in $\mathcal{Y}$, i.e. a zero-shot transfer learning task. For domain $\mathcal{X}$ we use the MNIST handwritten digits data set [24], and for domain $\mathcal{Y}$ we use the proposed BlockStacks data set, consisting of a single stack of multi-colored cubes of differing heights, each containing one randomly-positioned red cube (see Appendix B for details and examples). The shared signature includes the ordinal relations $\sigma = \{\mathsf{G}, \mathsf{E}, \mathsf{L}, \mathsf{S}, \mathsf{P}\}$, and it is applied to digit ordering in MNIST and to red cube position ordering in BlockStacks. We provide results with respect to a theory of ordinality, as explored in Example 1. A formal specification of the theory is provided in Appendix F. When transferring relations from $\psi_{\mathcal{X}}^{\text{enc}}$ to $\psi_{\mathcal{Y}}^{\text{enc}}$, one could use the full set $\phi$ of relation-decoders. However, this is not necessary from a logical standpoint because the entire system of relations can be expressed in terms of the isSuccessor relation $\mathsf{S}$ (e.g. the successor of a number is larger than that number). We therefore only employ the isSuccessor relation-decoder as the fixed-parameter selection to guide the learning of $\psi_{\mathcal{Y}}^{\text{enc}}$. If coherence, as defined in this paper, is carried across domains, we would expect the transferring of isSuccessor to produce an improved performance on the remaining relations in the target domain.

**Neural model components and hyperparameters:** Together with DC, existing relation-decoder models evaluated here are: TransR [25], HolE [30], NTN [41]. We additionally include a basic feedforward neural network (NN). To produce domain-encodings, all experiments use a $\beta$-VAE [17]. We provide further details for all models, including training regimen, parameterization and implementation in Appendix D. In the source domain, we explore $\beta$ values in $\{1, 4, 8, 12\}$ and set $\lambda = 10^3$. In the target domain, we first normalise losses and set $\beta = 10^{-4}$ and $\lambda = 10^{-2}$, as these produced good image reconstructions while optimising $L^{\tilde{\mathcal{Y}}_\sigma}$. In all experiments we fix $\mathcal{Z} = \mathbb{R}^{10}$.

## 7 Experimental Results and Discussion

In this section, experimental results show that transfer learning performance is positively correlated with our measures for consistency and coherence. This holds particularly true for embeddings that are different but near in space to source domain embeddings. As we have argued, for a neural model to perform well on concept transfer, its representations must maintain high probability of consistency with a theory that provides a semantics for the concept. The most robust way of doing this is to maintain consistency across regions of embedding space, rather than relying exclusively on the specific data-points observed at training time in the source domain. In our analysis, consistency losses are evaluated when sampling from different regions of latent space $\mathcal{Z}$. We evaluate: *data-embeddings*, where all inputs are encodings of a domain's test data; *interpolation*, when we derive an

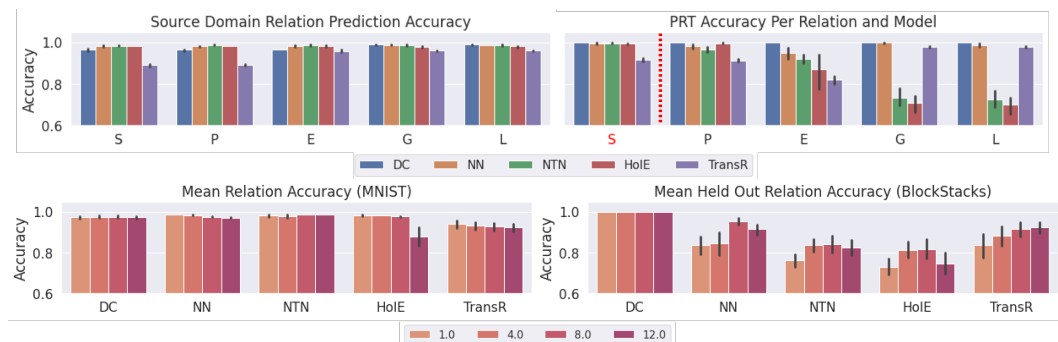

Figure 2: **[Top]** Relation-decoder prediction accuracy per model (DC, NN, NTN, HoIE, TransR) and relation (abbreviated on the $x$-axis as in Example 1), in the source domain (MNIST, left) and target domain (BlockStack, right). A red highlighted S and dotted line (top right) indicates that relation isSuccessor is included in training the target domain autoencoder, but none of the other relations are. Both DC and NN retain a good performance while all other models show a decrease of accuracy in the target domain for one or more of the relations not included in training. **[Bottom]** Impact of different values of $\beta \in \{1, 4, 8, 12\}$ for each relation-decoder averaged across all relations in the source domain (left) and held-out relations $\{P, E, G, L\}$ in the target domain (right). It can be seen that DC is not impacted by changes in $\beta$ and it maintains performance in the target domain. All other models show a decrease of accuracy for the held-out relations in the target domain.

empirical mean and variance for the domain's data-embeddings and sample from a corresponding Gaussian distribution; and *extrapolation*, when we sample from regions strictly outside the smallest, axis-aligned, hyper-rectangle that encloses all data-points.

Figure 2-top provides relation-decoder prediction accuracies in both the source (MNIST, left) and target (BlockStacks, right) domains.[7] The relations are S, P, E, G, L and relation S is transferred to the target domain. Key observations are that DC produces excellent PRT performance, whilst NN, NTN and HoIE all see some degradation from their source-domain accuracies for relations other than isSuccessor (S). TransR maintains target-domain accuracies similar to its performance in the source domain, but this is significantly below the performance of other models in the source domain. We include the impact of adjusting $\beta$ (disentanglement pressure) in Figure 2-bottom. Barring DC which has little discernible change in either source or target domains, PRT performance is significantly impacted by $\beta$ for all models in the target domain, but has little effect in the source domain. TransR shows a strong positive correlation between target domain accuracy and $\beta$ values, whereas the remaining models produce their best PRT performance with medium disentanglement pressure.

To investigate the broad trends that run across all $\beta$ values and relation-decoder models, we ran a Spearman rank correlation analysis between consistency losses and PRT performance. Separate coefficients are produced for each combination of consistency loss: Con-A and Con-I further divided into Con-I-T (transitivity), Con-I-A (asymmetry) and Con-I-R (reflexivity), and regions of latent space: data-embedding, interpolation and extrapolation (see Appendix H for the full table). Lower consistency losses are expected to produce higher PRT performance as indicated by a negative Spearman rank coefficient. The coefficients show that the consistency losses of data-embeddings in the source domain are weakly rank correlated with PRT. The consistency losses in the case of interpolation are, in most cases, strongly rank correlated with PRT. The consistency losses in the case of extrapolation lie in between and are generally moderately rank correlated with PRT. This supports our thesis that consistency can facilitate reliable transfer. Furthermore, consistency of certain partial theories may matter more. Here, Con-A, Con-I-T and Con-I-A on interpolation are the most relevant partial theories for transfer learning performance. As we shall see, DC outperforms all other models on these losses and this result is mirrored by its PRT performance.

To gain a deeper insight as to which underlying characteristics can explain the observed PRT accuracy profiles, Figure 3 and Figure 4 present Con-A and Con-I loss profiles, respectively, for varied $\beta$ and regions of latent space (for data-embeddings in blue, interpolation in green and extrapolation in

---

[7]We take $\phi_r$ inferences of 0.5 or above to signify *true*, and otherwise *false*. An alternative, left as future work, would be to sample the space of $\phi$ values to produce a confidence measure.

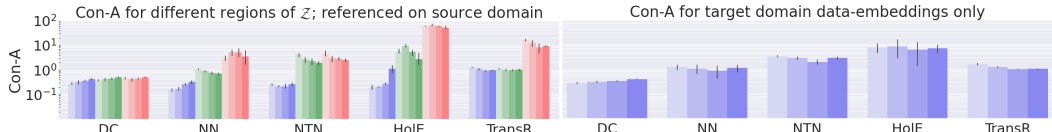

Figure 3: Consistency-Across (Con-A) losses (lower values are better) for the models (DC, NN, NTN, HolE, TransR) using the MNIST data set (source domain $\mathcal{X}$) **[left]** and BlockStacks (target domain $\mathcal{Y}$) **[right]**. The blue bars show the consistency loss of the data embeddings, with darker shades corresponding to models trained with higher $\beta$ (disentanglement pressure). The green bars show the results for interpolation. The red bars show the results for extrapolation.

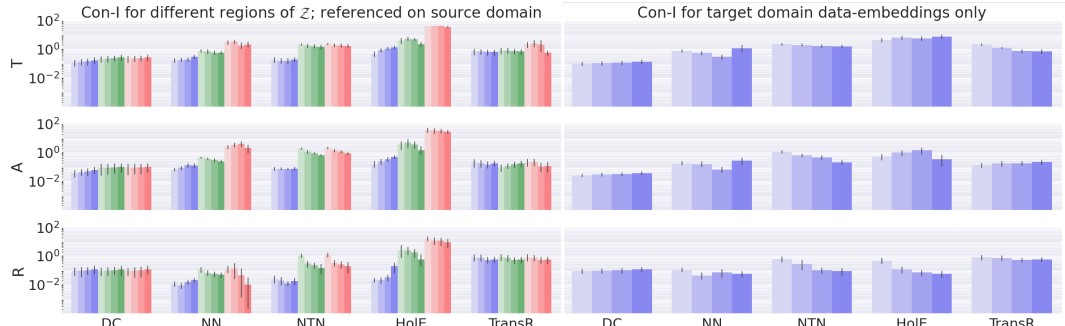

Figure 4: Consistency-Individual (Con-I) losses (lower values are better) for the models (DC, NN, NTN, HolE, TransR) using the MNIST data set (source domain $\mathcal{X}$) **[left]** and BlockStacks (target domain $\mathcal{Y}$) **[right]**. From top to bottom (following the same colour schematic as Figure 3): Con-I-T (**T**ransitivity), Con-I-A (**A**symmetry) and Con-I-R (**R**eflexivity).

red). Results refer to both source (left) and target domain embeddings (right). Firstly, we note that DC retains excellent Con-A for all regions of latent space. TransR retains consistency from data-embeddings to the interpolated regions, but not to the extrapolated regions. The remaining models show degradation of consistency between data-embeddings and interpolation and extrapolation regions, with extrapolation often being worse than interpolation. Looking at $\beta$ trends, aside from DC, increasing $\beta$ appears to have a positive but limited effect on interpolation and extrapolation performance. Considering the Con-A performance of data-embeddings in the target domain, DC shows the best performance. The Con-A performance in the target domain is in agreement with the PRT accuracies. For all the models, Con-A performance in the target domain appears to match the interpolation or extrapolation Con-A performance in the source domain. This points to the possibility of anticipating transfer learning performance by evaluating the consistency of partial theories.

Many of the same trends can be seen in the results for Con-I (Con-I-T, Con-I-A and Con-I-R) in Figure 4. Results are averaged over individual relations. As in Figure 3, results are presented with respect to source domain (left) and target domain (right). We firstly observe that DC and NN share the best overall Con-I performance profiles, with TransR following closely. DC and TransR again show comparable data-embedding versus interpolation/extrapolation performance, whereas NN, NTN and HolE suffer from degradation. With regards to $\beta$'s impact, DC is not affected by $\beta$, while NN and NTN show a negative correlation between $\beta$ and Con-I losses with comparable results for each underlying partial theory.

Finally, Table 1 provides a comparison between optimal coherences achieved for each relation-decoder model, as defined in Section 4. Results are partitioned according to each consistency type and aggregate value. DC clearly outperforms all other models on coherence. NN achieves strong aggregate coherence, followed by TransR and NTN, with HolE performing generally worse. Looking at $\beta^*$ profiles, we see that most models achieve optimum aggregate coherence at $\beta = 8$, apart from DC and HolE which perform better at $\beta = 1$. Overall, this is in broad agreement with the $\beta$ profiles given by Figure 2-bottom (right). However, we can see that $\beta^*$ profiles for Con-A coherence are in more direct agreement as TransR achieves its best at $\beta = 12$ and HolE at $\beta = 8$. This suggests that

Table 1: Coherence comparison with respect to source and target data-embeddings. Results are reported with the corresponding $\beta = \beta^*$ value (in parenthesis). The consistency loss abbreviations refer to: (A)cross, T(ransitivity), A(symmetry), R(eflexivity) and Aggr(egate), which gives the best obtained aggregate consistencies. DC outperforms all other approaches across most coherence scores.

| $\phi$ | Aggr. | $(\beta^*)$ | Con-A | $(\beta^*)$ | Con-I-T | $(\beta^*)$ | Con-I-A | $(\beta^*)$ | Con-I-R | $(\beta^*)$ |
|--------|-------|-------------|-------|-------------|---------|-------------|---------|-------------|---------|-------------|
| HolE | 12.12 | (1) | 6.61 | (8) | 4.30 | (1) | 0.52 | (12) | 0.08 | (8) |
| NTN | 4.11 | (8) | 1.92 | (8) | 1.50 | (12) | 0.22 | (12) | 0.09 | (12) |
| TransR | 2.51 | (8) | 1.02 | (12) | 0.71 | (12) | 0.18 | (4) | 0.55 | (8) |
| NN | 1.71 | (8) | 0.82 | (8) | 0.44 | (12) | 0.18 | (4) | **0.05** | (4) |
| DC | **0.53** | (1) | **0.29** | (1) | **0.11** | (1) | **0.04** | (1) | 0.09 | (1) |

Con-A is more indicative of PRT performance, which is to be expected since PRT relies on inductive transfer across relations.

All in all, these results paint a picture where source domain accuracy alone is not a strong enough indicator of concept transfer. Instead, it may be possible to anticipate transfer performance by evaluating consistency in regions beyond the source domain's data-embeddings. Depending on the task at hand, certain partial theories may be more relevant than others in this analysis.

## 8 Related Work

Relational representations play a prominent role in Knowledge Graph Embedding (KGE), wherein sets of relation-decoders are jointly learned to obtain a semantic latent representation from data [41, 45, 44, 5, 29, 47, 11, 20, 1, 38, 13, 3]. Although KGE approaches typically do not use a shared autoencoder as done in this paper, in [37] an autoencoding framework is adopted, where a graph neural network is used as the encoder. However, [37] did not work with visual data and the model was only applied to single data sets rather than transfer learning. Similarly, disentanglement is concerned with semantic representation learning [4], and it has been explored using a variety of methods including both Generative Adversarial Networks [10] and VAEs [6, 17, 9, 36, 14, 22, 26]. Disentangled representations have been evaluated on their transferability [46, 42, 27]. A bridge between these two fields, with relation-decoders employed in the semi-supervision of VAEs, can be found in [19, 8, 7]. In [19], multiple relation-decoders are used, but to compute a triplet comparison-based query. In [8, 7], only a single binary relation is studied using functional forms that are not sufficient to model the full set of relations considered in this paper. Lastly, we note that our experimental setup is most remnant of domain adaptation, e.g. [35]. To the best of our knowledge, this paper is the first to present a comprehensive analysis of the resulting concept coherence. No previous work has compared relation-decoders on their ability to learn consistently and coherently, as measured in this paper.

## 9 Conclusion and Future Work

This paper introduced formal definitions of consistency and coherence for representation learning. As a result, a sub-symbolic model can have its consistency and coherence measured with respect to a logical theory. The paper specified a neuro-symbolic model based on domain-encoders coupled with modular relation-decoders, and an experimental procedure that, together, allowed for the investigation of how concept coherence differs for various implementations of relation-decoders applied to transfer learning. Finally, consistency and coherence results showed that the models that can retain consistency (*i.e.* be coherent) across regions of latent space beyond the source data-embeddings are more likely to perform better at PRT learning tasks. The empirical evaluations in this paper only considered binary relations and a fixed signature which is learned "all at once" in a source domain. In practical applications, however, it should be possible to discover concepts gradually, e.g. as part of a curriculum and through gradual refinement of pre-learned relations after exposure to different contexts. This necessitates an adaptation of the approach presented here and further evaluations, as part of future work. Further evaluations of the formalization introduced here should consider the use of different models, theories (such as specifying periodic, *e.g.* rotation, and unordered categorical, *e.g.* shape, properties) and scenarios/data sets in the evaluation of consistency and coherence of neural models.

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
