

Figure 5: Example of two BlockStacks data set images. Each instance consists of a single red block varying in position within the block stack. On the left the red block is at height 3 (using a zero index) and on the right it is at height 1.

## A  Societal Impact Statement

This work does not have a negative societal impact, specifically it does not include any of the following: involvement of human subjects, sensitive data, harmful insights, methodologies and applications. The results, data sets and methodologies are objectively non-discriminatory, unbiased and fair. This work does not breach any privacy or security guidelines or laws, nor any other legal restrictions.

The proposed definition of coherent concepts and corresponding analysis provides more depth in the assessment of deep learning methods, which are typically otherwise opaque, and this can have a positive societal impact. Currently, we cannot provide interpretable descriptions regarding *how* a standard deep learning method produces its inferences, making it difficult to fully trust a model in critical applications. An important failure case is that biases are not easy to uncover from a trained deep learning model. The benefit of learning a coherent concept is that inferences uphold logical consistency, which can be formally expressed and tested. This can provide more trust in the model as practitioners can have confidence that the model should not obtain inputs that lead to incoherent inferences, wherein errors are certain. Further, if the logic does not include biases, the inferences of a coherent set of relation-decoders should not be biased. A caveat to these points is that unless the relation-decoder functional form allows us to analytically make comments/assertions about the model's performances for arbitrary regions of latent space, as with DC (see D.1), it is intractable to fully examine model coherence, as it requires a full extrapolation/interpolation evaluation. Nonetheless, a practical evaluation of coherence is an important step forward.

## B  BlockStacks dataset description

The BlockStacks data set consists of 12,000 RGB images ($3 \times 200 \times 200$ pixels but resized in code to $3 \times 128 \times 128$) of individual block stacks, of varying height (between 1-10 blocks), block colors (uniformly sampled from options: {gray, blue, green, brown, purple, cyan, yellow}) and position (uniformly sampled from $x, y$ range (-3,-3) to (3,3)), but with the requirement that each instance consists of a single red block at a random height (see Figure 5 for example images). These were rendered using the CLEVR rendering agent with the help of code from [2]. The dataset is divided into 9000:1500:1500 train, validation and test splits.

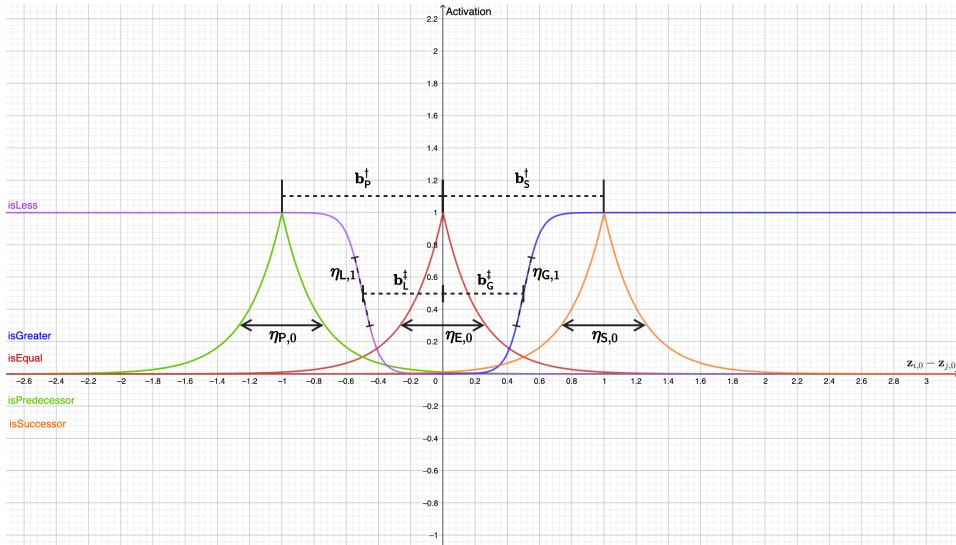

Figure 6: Depiction of a set of DC relation-decoders for binary relations isGreater, isLess, isEqual, isSuccessor and isPredecessor. Each DC relation-decoder (for each relation) shown here has a one-hot mask, $\boldsymbol{u}_r$, that is in this example the same across relations, which ensures only the zeroth dimensions of the embedding arguments are compared, giving $\boldsymbol{z}_{i,0}$ and $\boldsymbol{z}_{j,0}$.

## C  Explanation of the $\beta$-VAE

The VAE is derived by introducing an approximate posterior $q_\alpha(\boldsymbol{Z}|\boldsymbol{X})$, from which a lower bound (commonly referred to as the Evidence LOwer Bound (ELBO)) on the true marginal $\log p_\theta(\boldsymbol{X})$ can be obtained by using Jensen's inequality [21]. The VAE maximises the log-probability by maximising this lower bound, given by:

$$L_{\beta\text{-VAE}}^{\text{ELBO}} = \mathbb{E}_{q_\alpha(\boldsymbol{Z}|\boldsymbol{X})}[\log p_\theta(\boldsymbol{X}|\boldsymbol{Z})] - \beta D_{KL}(q_\alpha(\boldsymbol{Z}|\boldsymbol{X})\|p_\theta(\boldsymbol{Z})), \tag{15}$$

where $q_\alpha(\boldsymbol{Z}|\boldsymbol{X})$ is typically modelled as a neural-network encoder with parameters $\alpha$. Similarly $p_\theta(\boldsymbol{X}|\boldsymbol{Z})$ is often modelled as a neural-network decoder with parameters $\theta$ and is calculated as a Monte Carlo estimation. A reparameterization trick is used to enable differentiation through an otherwise undifferentiable sampling from $q_\alpha(\boldsymbol{Z}|\boldsymbol{X})$ (see [21]). In the $\beta$-VAE [17, 6], an additional $\beta$ scalar hyperparameter was added as it was found to influence disentanglement through stronger distribution matching pressure with respect to the prior $p_\theta(\boldsymbol{Z})$, where this prior is typically set to an isotropic zero-mean Gaussian $\mathcal{N}(\boldsymbol{0}, \boldsymbol{I})$). When $\beta = 1$ we obtain the standard VAE objective [21].

## D  Model Descriptions

In this section we firstly present an in-depth analysis of the key innovations presented by DC which provides insight into how it can learn a coherent notion of ordinality. We then provide model details for each of the compared relation-decoders in the main results and the backbone $\beta$-VAE architecture that we employ for each data set.

### D.1  Dynamic Comparator Analysis

Figure 6 depicts how DC is able to learn the isGreater, isLess, isEqual, isSuccessor and isPredecessor family of binary ordinal relations, assuming each corresponding relation-decoder has learned a common one-hot mask on the zeroth dimension *i.e.* $\boldsymbol{u}_\mathsf{G} = \boldsymbol{u}_\mathsf{E} = \ldots = \boldsymbol{u}_\mathsf{P} = [1, 0, \ldots, 0]$, such that activations only depend on the $\boldsymbol{z}_{i,0} - \boldsymbol{z}_{i,1}$ difference. An important capability of DC is its ability to dynamically *select* via $\boldsymbol{a}_r$ an appropriate functional mode, either $\phi_r^\dagger$ or $\phi_r^\ddagger$, depending on the type of relation it needs to model. As shown by Figure 6, this allows isEqual to exhibit its reflexive, symmetric and transitive characteristics, whilst isGreater and isLess both carry transitivity but are asymmetric and irreflexive. Furthermore, the use of a subtraction between $\boldsymbol{z}_i$ and $\boldsymbol{z}_j$ (which, via mask

$u$, ends up only being a subtraction between their zeroth dimensions) leads to a relative comparison, not an absolute comparison, which generalises to arbitrary $z_i$ and $z_j$ sampled from anywhere in $\mathcal{Z}$.

Note that there is no built in parameter sharing, meaning each relation-decoder (for each individual relation $r$) is trained independently and has its own set of $a_r, u_r, \eta_{r,0}, \eta_{r,1}, b_r^{\dagger}$ and $b_r^{\ddagger}$ parameters. However, our experiments show that DC reliably obtains settings such that $e.g.$ $u_{\mathsf{G}} = u_{\mathsf{E}}$, or $a_{\mathsf{G}} = a_{\mathsf{L}} = [0,1]$, or $b_{\mathsf{G}}^{\ddagger} = -b_{\mathsf{L}}^{\ddagger}$ and so on. DC is thus able to discover the interdependencies between families of relations. By learning to loosely 'tie' together parameters in this way, whilst still being expressive enough to model each type of relation, DC can facilitate a data-driven binding between relation-decoder outputs. This helps ensure consistent generalisation across a latent subspace, as defined by the common/overlapped $u_r$ masks.

### D.2 Relation-Decoder implementations

**TransR [25]:**
$$\phi_r^{\text{TransR}}(z_i, z_j) = \|h_r + r - t_r\|_2^2$$
with,
$$h_r = M_r z_i \quad \text{and} \quad t_r = M_r z_j.$$
where for $z_i, z_j \in \mathbb{R}^{d_z}$ vectors, $M_r \in \mathbb{R}^{d_z \times d_z}$ and $r \in \mathbb{R}^{d_z}$. As we want to obtain a $(0,1)$ output, we modify TransR through $\phi_r^{\text{TransR}^+} = \sigma(c - \phi_r^{\text{TransR}})$, where $\sigma$ is a sigmoid function and c is a scalar that ensures that at $\phi_r^{\text{TransR}}(z_i, z_j) = 0$, then $\phi_r^{\text{TransR}^+}(z_i, z_j) \approx 1$. In all experiments we set $c = 10$.

**NTN** (modified version of [41] from [13, 38]):
$$\phi_r(z_1, \ldots, z_n) = \sigma\big(u_r^{\top}[\tanh(z^{c\top} M_r z^c + V_r z^c + b_r)]\big)$$
$$\tag{16}$$

where $u_r \in \mathbb{R}^k, M_r \in \mathbb{R}^{n \cdot d_z \times n \cdot d_z \times k}, V_r \in \mathbb{R}^{k \times n \cdot d_z)}$ and $b_r \in \mathbb{R}^k$. The only hyperparameter to consider is $k$, which controls the NTN's capacity - in all experiments, we set this to 1. If $k > 1$, $z^{c\top} M_r z^c$ produces a $k$-dimension vector by applying the bilinear operation to each of the $k$ $M_r$ slices. Here $z^c \in \mathbb{R}^{n \cdot d_z}$ is a concatenation of the inputs $z_1, \ldots, z_n$, which was introduced in [13, 38]. In contrast, the original NTN (see [41]) is only applicable to binary relations and does not include the outer sigmoid.

**HolE [30]:**
$$\phi_r^{\text{HolE}}(z_i, z_j) = \sigma\big(r^{\top}(z_i \star z_j)\big)$$
where $r \in \mathbb{R}^{d_z}$ and $\star : \mathbb{R}^{d_z} \times \mathbb{R}^{d_z} \to \mathbb{R}^{d_z}$ denotes the circular correlation operator and is given by,
$$[z_i \star z_j]_k = \sum_{m=0}^{d-1} z_{i,m} z_{j,(k+m) \mod d}$$

**NN**: a simple four-layer neural-network with layer sizes $l_{\text{in}} = 2d_z, l_1 = 2d_z$ and $l_2 = d_z$, with ReLU activations [28]. The final output layer, $l_{\text{out}}$, is a single value passed through a sigmoid function, to bound the output within $(0,1)$.

### D.3 $\beta$-VAE configuration

The model configurations used for both MNIST and BlockStacks data sets are given in Table 2.

### D.4 $L^{joint}$ configuration

In the source domain, we vary $\beta$ values between $\{1, 4, 8, 12\}$ and fix $\lambda = 10^3$. In the target domain, we fix $\beta$ to $10^{-4}$ and $\lambda = 10^{-2}$ and normalise the $\mathcal{L}_{\beta\text{-VAE}}^{ELBO}$ reconstruction term by dividing by a factor $\frac{1}{\sqrt{H \cdot W \cdot C}}$, for height $H$, width $W$ and color channels $C$, and normalize the distribution matching term by a factor $\frac{1}{d_z}$, for latent representation size $d_z$ (set to 10 across all experiments).

To train relation-decoders over a given domain $\mathcal{S}$, it is necessary to supervise estimates of $\phi_r(\psi_{\mathcal{S}}^{enc}(O)), O \in \mathcal{S}^2$, against corresponding ground-truth labels, $\gamma_{O,\mathcal{S}_{\sigma}}^r$. However, doing so for

Table 2: Specification of our $\beta$-VAE encoder and decoder model parameters, for both 28×28 (top) and 128×128 (bottom) size input data. I: Input channels, O: Output channels, K: Kernel size, S: Stride, P: Padding, A: Activation

**Encoder**
Input: $28 \times 28 \times N_C = 1$

| **Layer_ID ; I ; O ; K ; S ; P ; A** |
| --- |
| Conv2d_1 ; $N_C$ ; 32 ; $4 \times 4$ ; 2 ; 1 ; ReLU |
| Conv2d_2 ; 32 ; 32 ; $4 \times 4$ ; 2 ; 1 ; ReLU |
| Conv2d_3 ; 32 ; 64 ; $3 \times 3$ ; 2 ; 1 ; ReLU |
| Conv2d_4 ; 64 ; 64 ; $2 \times 2$ ; 2 ; 1 ; ReLU |

| **Layer_ID ; Num Nodes : In - Out ; A** |
| --- |
| FC_z ; 576 - 144 ; ReLU |
| FC_z_mu ; 144 - 10 ; None |
| FC_z_logvar ; 144 - 10 ; None |

**Decoder**
Input: $\mathbb{R}^{10}$

| **Layer_ID ; Num Nodes : In - Out ; A** |
| --- |
| FC_z ; 10 - 144 ; ReLU |
| FC_z_mu ; 144 - 576 ; ReLU |

| **Layer_ID ; I ; O ; K ; S ; P ; A** |
| --- |
| UpConv2d_1 ; 64 ; 64 ; $2 \times 2$ ; 2 ; 1 ; ReLU |
| UpConv2d_2 ; 64 ; 32 ; $3 \times 3$ ; 2 ; 1 ; ReLU |
| UpConv2d_3 ; 32 ; 32 ; $4 \times 4$ ; 2 ; 1 ; ReLU |
| UpConv2d_4 ; 32 ; $N_C$ ; $4 \times 4$ ; 2 ; 1 ; Sigmoid |

**Encoder**
Input: $128 \times 128 \times N_C = 3$

| **Layer_ID ; I ; O ; K ; S ; P ; A** |
| --- |
| Conv2d_1 ; $N_C$ ; 32 ; $4 \times 4$ ; 2 ; 1 ; ReLU |
| Conv2d_2 ; 32 ; 32 ; $4 \times 4$ ; 2 ; 1 ; ReLU |
| Conv2d_3 ; 32 ; 64 ; $4 \times 4$ ; 2 ; 1 ; ReLU |
| Conv2d_4 ; 32 ; 64 ; $4 \times 4$ ; 2 ; 1 ; ReLU |
| Conv2d_5 ; 64 ; 64 ; $4 \times 4$ ; 2 ; 1 ; ReLU |

| **Layer_ID ; Num Nodes : In - Out ; A** |
| --- |
| FC_z ; 1024 - 256 ; ReLU |
| FC_z_mu ; 256 - 10 ; None |
| FC_z_logvar ; 256 - 10 ; None |

**Decoder**
Input: $\mathbb{R}^{10}$

| **Layer_ID ; Num Nodes : In - Out ; A** |
| --- |
| FC_z ; 10 - 256 ; ReLU |
| FC_z_mu ; 256 - 1024 ; ReLU |

| **Layer_ID ; I ; O ; K ; S ; P ; A** |
| --- |
| UpConv2d_1 ; 64 ; 64 ; $4 \times 4$ ; 2 ; 1 ; ReLU |
| UpConv2d_2 ; 64 ; 32 ; $4 \times 4$ ; 2 ; 1 ; ReLU |
| UpConv2d_3 ; 32 ; 32 ; $4 \times 4$ ; 2 ; 1 ; ReLU |
| UpConv2d_4 ; 32 ; 32 ; $4 \times 4$ ; 2 ; 1 ; ReLU |
| UpConv2d_5 ; 32 ; $N_C$ ; $4 \times 4$ ; 2 ; 1 ; Sigmoid |

every $O \in \mathcal{S}^2$ can easily become intractable and we instead only sample a subset of possible $\mathcal{S}^2$ tuples. Our sampling strategy involves first selecting a ratio $R = \frac{|\mathcal{B}|}{|\mathcal{S}|}$ where $\mathcal{B} \subset \mathcal{S}^2$ is a set of $O$ tuples. We then sample relation-decoder specific subsets $\mathcal{B}_r$ where $|\mathcal{B}_r| = \frac{|\mathcal{B}|}{|\sigma|}$, to ensure a balanced distribution of tuples between relation-decoders. Furthermore, we ensure that each $\mathcal{B}_r$ contains a balanced ratio of $\gamma^r_{O,\mathcal{S}_\sigma} = 1$ versus $\gamma^r_{O,\mathcal{S}_\sigma} = 0$ instances. We found that each $|\mathcal{B}_r|$ set can be small without jeopardising the final relation-decoder performance level, allowing us to use $R = 1$ for MNIST experiments and $R = 3$ for BlockStacks experiments.

Finally, in all experiments we use a $\beta$-VAE trained for up to 300,000 steps, following accepted practice from [26, 42], together with any included relation-decoders. However, to ensure computation efficiency across experiments, we employ an early stopping procedure, where if the validation score does not increase over 30 and 120 training epochs for MNIST and Blockstacks experiments, respectively, we end the training early.

# E  Preliminaries in further detail

**Logic and model-theoretic background:** to support Section 2 we provide additional logic and model theoretic background. In this paper, we assume a formal language $\mathcal{L}$ composed of variables, predicates (i.e. relations), logical connectives $\neg$ (negation), $\vee$ (disjunction), $\wedge$ (conjunction), $\rightarrow$ (implication), and universal quantification $\forall$ (for all) with their conventional meaning (see [39]). The set of relations in $\mathcal{L}$ form the *signature*, $\sigma$, of the language. Relations have an associated arity, denoted as $\mathrm{ar}(\cdot)$, that defines the number of arguments they take. For example, a binary relation $r$ has arity $\mathrm{ar}(r) = 2$. Relations are used to express knowledge over the elements of a *domain* $\mathcal{S}$, where $\mathcal{S}$ is a non-empty set. For instance, $r(s_1, s_2)$ states that elements $s_1$ and $s_2$ are related through the binary relation $r$. The meaning of a relation is defined by an *interpretation* $I_{\mathcal{S}_\sigma}$ which captures

the $\{T, F\}$ (true or false) values of the relation over elements of $\mathcal{S}$. Together, a domain $\mathcal{S}$ and an interpretation $I_{\mathcal{S}_\sigma}$ of a given signature $\sigma$ form a *structure* $\mathcal{S}_\sigma = (\mathcal{S}, I_{\mathcal{S}_\sigma})$.

Note that for a fixed domain $\mathcal{S}$ and signature $\sigma$, different interpretations yield different structures. As stated in the main text, we construct universally quantified first-order formulae (called sentences) using the signature $\sigma$ of $\mathcal{L}$, whose truth-value is defined with respect to a given structure $\mathcal{S}_\sigma$. To do so, we first consider *ground* instances of a formula. These are given by replacing all the variables in the formula with elements from the domain $\mathcal{S}$. For example, $r(s_1, s_2)$, where $s_1$ and $s_2$ are elements of $\mathcal{S}$, is a *ground* instance of an atomic formula $r(i, j)$ where $i$ and $j$ are variables in $\mathcal{L}$. Given a structure $\mathcal{S}_\sigma = (\mathcal{S}, I_{\mathcal{S}_\sigma})$, a relation $r$, and a tuple $(s_1, \ldots, s_{\mathsf{ar}(r)}) \in \mathcal{S}^{\mathsf{ar}(r)}$, a ground instance $r(s_1, \ldots, s_{\mathsf{ar}(r)})$ is true in the structure $\mathcal{S}_\sigma$ if and only if $(s_1, \ldots, s_{\mathsf{ar}(r)}) \in I_{\mathcal{S}_\sigma}(r)$. The truth value of a sentence in a given structure $\mathcal{S}_\sigma$ depends on the truth value of its respective ground instances. Specifically, a sentence is true in a structure $\mathcal{S}_\sigma$ if and only if all of its ground instances are true in $\mathcal{S}_\sigma$. For example, $\forall i. r(i, i)$ is true in $\mathcal{S}_\sigma$ if and only if all of its ground instances $r(s_h, s_h)$ are true in $\mathcal{S}_\sigma$, for every $s_h \in \mathcal{S}$. When a sentence, $\tau$, is true in a structure, $\mathcal{S}_\sigma$, we say that the structure *satisfies* $\tau$, denoted as $\mathcal{S}_\sigma \models \tau$. A set of sentences form a *theory*, $\mathcal{T}$ and any subset of the sentences in $\mathcal{T}$ form a partial theory with respect to $\mathcal{T}$. A theory can be seen as a way of constraining the type of interpretations that we want to "accept" for our signature. Finally, a *model* of $\mathcal{T}$ is a structure that satisfies every sentence in $\mathcal{T}$.

# F  Specification for theory of ordinality

To support our claim that we can use only the isSuccessor relation as the target encoder guide due to its logical relationship with the remaining relations, we include here the logical clauses:

$$\forall i, j, k. \ (\mathsf{isSuccessor}(i, j) \wedge \mathsf{isSuccessor}(k, j) \rightarrow \mathsf{isEqual}(i, k))$$
$$\forall i, j. \ (\mathsf{isSuccessor}(i, j) \rightarrow \mathsf{isGreater}(i, j))$$
$$\forall i, j, k. \ (\mathsf{isSuccessor}(i, j) \wedge \mathsf{isGreater}(j, k) \rightarrow \mathsf{isGreater}(i, k))$$
$$\forall i, j. \ (\mathsf{isSuccessor}(i, j) \leftrightarrow \mathsf{isPredecessor}(j, i))$$
$$\forall i, j. \ (\mathsf{isPredecessor}(i, j) \rightarrow \mathsf{isLess}(i, j))$$
$$\forall i, j, k. \ (\mathsf{isPredecessor}(i, j) \wedge \mathsf{isLess}(j, k) \rightarrow \mathsf{isLess}(i, k)).$$

Therefore, by knowing all of the successor relations between data instances, it should be possible to infer the remaining relationships that they share.

For completeness, we provide the truth tables for each of the sub-theories that our consistency losses evaluate against. We only include configurations that are valid under the constraints, indicated by $\subset \mathcal{T} = T$, where this notation highlights the fact each incomplete set of constraints form a subset of the overall theory $\mathcal{T}$.

Firstly, the truth-table that describes constraints shared between relation truth-values is given by the following, $\forall i, j$:

| $\mathsf{G}(i, j)$ | $\mathsf{E}(i, j)$ | $\mathsf{L}(i, j)$ | $\mathsf{S}(i, j)$ | $\mathsf{P}(i, j)$ | $\subset \mathcal{T}$ |
|---|---|---|---|---|---|
| $T$ | $F$ | $F$ | $F$ | $F$ | $T$ |
| $T$ | $F$ | $F$ | $T$ | $F$ | $T$ |
| $F$ | $T$ | $F$ | $F$ | $F$ | $T$ |
| $F$ | $F$ | $T$ | $F$ | $F$ | $T$ |
| $F$ | $F$ | $T$ | $F$ | $T$ | $T$ |

where we use the same relation abbreviations as in the main text results.

Next, we provide each of the three consistency individual (Con-I) truth-tables. These are referred to as being "individual" due to the fact that they describe constraints applied to the truth-state of a single

Table 3: Characteristic properties of ordinal relations.

| Relation | asymmetric | transitive | reflexive |
|----------|------------|------------|-----------|
| G | Y | Y | N |
| E | N | Y | Y |
| L | Y | Y | N |
| S | Y | N | N |
| P | Y | N | N |

relation. For transitivity, given by the rule *e.g.* $\mathsf{G}(i,j) \wedge \mathsf{G}(j,k) \to \mathsf{G}(i,k)$, we have that $\forall i, j$:

$$
\begin{array}{ccc|c}
\mathsf{G}(i,j) & \mathsf{G}(j,k) & \mathsf{G}(i,k) & \subset \mathcal{T} \\
\hline
F & F & F & T \\
F & F & T & T \\
T & F & F & T \\
T & F & T & T \\
F & T & F & T \\
F & T & T & T \\
T & T & T & T \\
\end{array}
\tag{17}
$$

For asymmetry, where $\mathsf{S}(i,j) \to \neg \mathsf{S}(j,i)$, we have $\forall i, j$:

$$
\begin{array}{cc|c}
\mathsf{S}(i,j) & \mathsf{S}(j,i) & \subset \mathcal{T} \\
\hline
F & F & T \\
T & F & T \\
F & T & T \\
\end{array}
\tag{18}
$$

.

Finally, for reflexivity, given by $\mathsf{E}(i,i) \to \top$ (in this case describing that an object is always equal to itself) we have $\forall i$:

$$
\begin{array}{c|c}
\mathsf{E}(i,i) & \subset \mathcal{T} \\
\hline
T & T \\
\end{array}
\tag{19}
$$

Truth-table matrices for each of the above truth-tables can be obtained by replacing $T$ with $1$ and $F$ with $0$. The full set of individual constraints that are applicable to each relation covered in this paper are given by Table 3.

## G   Expanded consistency loss derivation

In this section, we present the expanded justification for reporting $-\ln 1 - \bar{\epsilon}$ consistency and coherence as a proxy for $\epsilon$-consistency/coherence as defined in Section 3. For notational clarity, in the following we omit $\psi_\mathcal{S}$, such that $\phi_r(\psi_\mathcal{S}(O))$ is abbreviated to $\phi_r(O)$.

In the following, we make no assumptions about the sizes of domain $\mathcal{S}$, signature $\sigma$ and arities of each $r \in \sigma$. Further, we take $\mathcal{T}$ to be an arbitrary theory over $\sigma$ consisting of universally quantified formula, and the validity of each ground instances of atomic formula with respect to $\mathcal{T}$, can be expressed by a single ground truth-table matrix, $\mathbf{T} \in \{0, 1\}^{K_0 \times K_1 \times K_2}$, wherein each slice, $\mathbf{T}_{k,:,:}$ gives a unique grounding of domain objects to the variables, $v$, required by $\mathcal{T}$. For each grounding of the $K_0 = |\mathcal{S}|^{|v|}$ possible groundings, there are $K_1 = 2^l$ unique truth-assignments to the $l$ atomic formulae that constitute $\mathcal{T}$, giving $K_2 = l + 1$ assignments per $\mathbf{T}_{k,t,:}$ row - one per atomic formulae and an additional value that denote whether the particular row satisfies $\mathcal{T}$. $\mathbf{T}$ can be obtained by taking any truth-table from the previous section and switching true (T) for 1 and false (F) for 0, and producing $K_0$ copies for each assignment of domain elements to the variables. Given this truth-table matrix, notice that a structure $\mathcal{S}_\sigma$ can be composed by selecting a single row of $\mathbf{T}$ for each grounding ($k$th slice), giving a vector $\boldsymbol{c}_{kt} = \mathbf{T}_{k,t,1:l}$. If the structure is a model of $\mathcal{T}$, *i.e.* $\mathcal{S}_\sigma \in \mathcal{M}_\mathcal{S}^\mathcal{T}$, then only rows with $\mathbf{T}_{k,t,K_2} = 1$ are allowed. Taking $t^+$ to be the set of rows such that $\mathbf{T}_{k,t,K_2} = 1$ (which is identical for each $k$) *i.e.* $t^+ = \{ t \,|\, \mathbf{T}_{k,t,K_2} = 1 \wedge t \in \{1, \ldots, K_1\}\}$, we can then rewrite $\Gamma_\mathcal{T}^{\mathcal{S}_\sigma}$ in

terms of samples from $\boldsymbol{T}$:

$$\Gamma_{\mathcal{T}}^{\tilde{\mathcal{S}}_\sigma} = \sum_{\mathcal{S}_\sigma \in \mathcal{M}_{\mathcal{S}}^{\mathcal{T}}} \prod_{r \in \sigma} \prod_{O \in \mathcal{S}^{\mathrm{ar}(r)}} \phi_r(O)^{\gamma_{O,\mathcal{S}_\sigma}^r} (1 - \phi_r(O))^{1 - \gamma_{O,\mathcal{S}_\sigma}^r} \qquad \text{(Eqn. 3)}$$

$$= \sum_{\mathcal{S}_\sigma \in \mathcal{M}_{\mathcal{S}}^{\mathcal{T}}} \prod_{k=1}^{K_0} \sum_{t \in t^+} \mathbf{1}_{t_k^{\mathcal{S}_\sigma}}(t) \prod_{m=1}^{l} f(\phi_{r^m}, O_{km}, c_{ktm})^{N(\phi_{r^m}, O_{km}, c_{ktm}, \mathcal{S}_\sigma)^{-1}} \qquad (20)$$

with

$$f(\phi_{r^m}, O_{km}, c_{ktm}) = \phi_{r^m}(O_{km})^{c_{ktm}} (1 - \phi_{r^m}(O_{km}))^{1 - c_{ktm}} . \qquad (21)$$

In the above, $\mathbf{1}_{t_k^{\mathcal{S}_\sigma}}(t)$ is an indicator function which equals 1 if $t = t_k^{\mathcal{S}_\sigma}$ and 0 otherwise, for active row $t_k^{\mathcal{S}_\sigma}$ under structure $\mathcal{S}_\sigma$ and grounding $k$. $\mathbf{1}_{t_k^{\mathcal{S}_\sigma}}(t)$ has the role of only including the *single* summand where $t$ corresponds with $t_k^{\mathcal{S}_\sigma}$. $N(\phi_{r^m}, O_{km}, c_{ktm}, \mathcal{S}_\sigma)$ is a function that counts the number of repeat products of term $f(\phi_{r^m}, O_{km}, c_{ktm})$, such that the appropriate root can be applied. We use $r^m$ to denote the relation for atomic formula at column $m$ and $O_{km}$ its corresponding arguments under grounding $k$; and we use $c_{ktm}$ to denote the truth-assignment of the atomic formula for column $m$, as designated by row $t$.

At this point, we are left with an expression for $\Gamma_{\mathcal{T}}^{\tilde{\mathcal{S}}_\sigma}$ in terms of truth-table matrix $\boldsymbol{T}$ entries, which is more reminiscent of $L(\mathcal{T}, \tilde{\mathcal{S}}_\sigma)$ as defined in Section 4. However, we must go further to expose the relationship between $\Gamma_{\mathcal{T}}^{\tilde{\mathcal{S}}_\sigma}$ and $L(\mathcal{T}, \tilde{\mathcal{S}}_\sigma)$ for arbitrary $\mathcal{T}$ expressed by $\boldsymbol{T}$. We will now show that the consistency loss $L(\mathcal{T}, \tilde{\mathcal{S}}_\sigma)$ gives the negative log-likelihood of satisfying $\mathcal{T}$ given a grounding $k \in \{1, \dots, K_0\}$, which can be further seen as a relaxation of $\Gamma_{\mathcal{T}}^{\tilde{\mathcal{S}}_\sigma}$ to sum over all rows $t \in t^+$ and without normalising via the $N(\phi_{r^m}, O_{km}, c_{ktm}, \mathcal{S}_\sigma)^{-1}$ exponent. With Boolean random variable $B_{\mathcal{T}}$ denoting whether $\mathcal{T}$ is ($b_{\mathcal{T}} = 1$) or is not ($b_{\mathcal{T}} = 0$) satisfied, the consistency loss for a soft-structure $\tilde{\mathcal{S}}_\sigma$ against theory $\mathcal{T}$ is given by,

$$L(\mathcal{T}, \tilde{\mathcal{S}}_\sigma) = \mathbb{E}_{k \sim U[\{1, \dots, K_0\}]}[H(p(B_{\mathcal{T}}|\mathcal{S}_\sigma, k), p(B_{\mathcal{T}}|\tilde{\mathcal{S}}_\sigma, k))] \qquad \text{Eqn. 8 base}$$

which can be expanded to,

$$L(\mathcal{T}, \tilde{\mathcal{S}}_\sigma) = -\sum_{k=1}^{K_0} \frac{1}{K_0} p(b_{\mathcal{T}} = 1|\mathcal{S}_\sigma, k) \ln p(b_{\mathcal{T}} = 1|\tilde{\mathcal{S}}_\sigma, k) \qquad (22)$$

$$+ (1 - p(b_{\mathcal{T}} = 1|\mathcal{S}_\sigma, k)) \ln(1 - p(b_{\mathcal{T}} = 1|\tilde{\mathcal{S}}_\sigma, k)).$$

where $\mathcal{S}_\sigma \in \mathcal{M}_{\mathcal{S}}^{\mathcal{T}}$. Given $\mathcal{S}_\sigma \in \mathcal{M}_{\mathcal{S}}^{\mathcal{T}}$, then $p(b_{\mathcal{T}} = 1|\mathcal{S}_\sigma, k) = 1$ always holds. This means the negative case in Eqn. 22 can be ignored, yielding the following simplified form:

$$L(\mathcal{T}, \tilde{\mathcal{S}}_\sigma) = -\sum_{k=1}^{K_0} \frac{1}{K_0} \ln p(b_{\mathcal{T}} = 1|\tilde{\mathcal{S}}_\sigma, k)$$

$$= -\mathbb{E}_{k \sim U[1, \dots, K_0]}[\ln p(b_{\mathcal{T}} = 1|\tilde{\mathcal{S}}_\sigma, k)]. \qquad \text{Eqn. 8}$$

and so $L(\mathcal{T}, \tilde{\mathcal{S}}_\sigma)$ is simply the negative log-likelihood of sampling a satisfied theory ($b_{\mathcal{T}} = 1$) from soft-structure $\tilde{\mathcal{S}}_\sigma$, for randomly sampled grounding $k$. Next, we show the similarities between $L(\mathcal{T}, \tilde{\mathcal{S}}_\sigma)$ and $\Gamma_{\mathcal{T}}^{\tilde{\mathcal{S}}_\sigma}$ by looking at the likelihood $p(b_{\mathcal{T}} = 1|\tilde{\mathcal{S}}_\sigma, k)$. First, we define $\bar{\Gamma}_{\mathcal{T}}^{\tilde{\mathcal{S}}_\sigma}$ by isolating the likelihood:

$$\exp(-L(\mathcal{T}, \tilde{\mathcal{S}}_\sigma)) = \prod_{k=1}^{K_0} p(b_{\mathcal{T}} = 1|\tilde{\mathcal{S}}_\sigma, k)^{\frac{1}{K_0}}$$

$$\doteq \bar{\Gamma}_{\mathcal{T}}^{\tilde{\mathcal{S}}_\sigma} \qquad (23)$$

We then expand $p(b_{\mathcal{T}} = 1|\tilde{\mathcal{S}}_\sigma, k)$ to:

$$p(b_{\mathcal{T}} = 1|\tilde{\mathcal{S}}_\sigma, k) = \sum_{t=1}^{K_1} p(b_{\mathcal{T}} = 1|\boldsymbol{c}_{kt}) p(\boldsymbol{c}_{kt}|\tilde{\mathcal{S}}_\sigma, k)$$

$$= \sum_{t \in t^+} p(\boldsymbol{c}_{kt}|\tilde{\mathcal{S}}_\sigma, k) \qquad (24)$$

Table 4: Spearman rank coefficients between consistency loss and PRT accuracy. Coefficients are calculated for each consistency loss reported in the main text, across all models, $\beta$ settings and regions of latent space. Results show a strong inverse rank correlation between interpolation Con-A/Con-I-T/Con-I-A and PRT performance.

| | Spearman Rank Coefficient | | | |
|---|---|---|---|---|
| $\mathcal{Z}$ region | Con-A | Con-I-T | Con-I-A | Con-I-R |
| Data-Embeddings | 0.2530 | -0.4451 | -0.4655 | 0.2307 |
| Interpolation | -0.7655 | -0.7479 | -0.7120 | -0.4233 |
| Extrapolation | -0.6005 | -0.6586 | -0.6140 | -0.4895 |

where $t^+$ is defined as before. For all other $t \neq t^+$, $p(b_{\mathcal{T}} = 1 | \boldsymbol{c}_{kt}) = 0$ and so this acts as a filter, yielding:

$$\bar{\Gamma}_{\mathcal{T}}^{\tilde{\mathcal{S}}_\sigma} = \prod_{k=1}^{K_0} \sum_{t \in t^+} p(\boldsymbol{c}_{kt} | \tilde{\mathcal{S}}_\sigma, k)^{\frac{1}{K_0}}. \tag{25}$$

$p(\boldsymbol{c}_{kt} | \tilde{\mathcal{S}}_\sigma, k)$ is calculated by evaluating the belief of each relation-decoder against the expected truth-assignment as defined by truth-table row $\boldsymbol{c}_{kt}$:

$$p(\boldsymbol{c}_{kt} | \tilde{\mathcal{S}}_\sigma, k) = \prod_{m=1}^{l} \phi_{r^m}(O_{km})^{c_{ktm}} (1 - \phi_{r^m}(O_{km}))^{1-c_{ktm}}$$
$$= f(\phi_{r^m}, O_{km}, c_{ktm})$$

where $r^m$ is the relation for atomic formula associated with column $m$ (which is the same for each $k$ slice and $t$ row) and $O_{km}$ is the grounding of this entry for slice $k$ (which is the same across rows). Putting it all back together, we finally have that:

$$\bar{\Gamma}_{\mathcal{T}}^{\tilde{\mathcal{S}}_\sigma} = \prod_{k=1}^{K_0} \sum_{t \in t^+} \prod_{m=1}^{l} f(\phi_{r^m}, O_{km}, c_{ktm})^{\frac{1}{K_0}}, \tag{26}$$

which makes the similarities between $\Gamma_{\mathcal{T}}^{\tilde{\mathcal{S}}_\sigma}$ and $\bar{\Gamma}_{\mathcal{T}}^{\tilde{\mathcal{S}}_\sigma}$ clear and exposes their relationship. In particular, for the special case where $|\mathcal{M}_{\mathcal{S}}^{\mathcal{T}}| = 1$, the outer sum for $\Gamma_{\mathcal{T}}^{\tilde{\mathcal{S}}_\sigma}$ can be removed, and the remaining differences between $\Gamma_{\mathcal{T}}^{\tilde{\mathcal{S}}_\sigma}$ and $\bar{\Gamma}_{\mathcal{T}}^{\tilde{\mathcal{S}}_\sigma}$ are the sum over $t^+$ rows and difference in exponent over $f(\phi_{r^m}, O_{km}, c_{ktm})$. For $\Gamma_{\mathcal{T}}^{\tilde{\mathcal{S}}_\sigma}$ to be maximised, through $p(\mathcal{S}_\sigma | \tilde{\mathcal{S}}_\sigma) \approx 1$, we would find that $\tilde{\mathcal{S}}_\sigma$ maximally supports only the rows associated with $\mathcal{S}_\sigma$ for each $k$ grounding. Notice that $\bar{\Gamma}_{\mathcal{T}}^{\tilde{\mathcal{S}}_\sigma}$ is again bound to $(0, 1)$ and achieves $\bar{\Gamma}_{\mathcal{T}}^{\tilde{\mathcal{S}}_\sigma} \approx 1$ when $\Gamma_{\mathcal{T}}^{\tilde{\mathcal{S}}_\sigma} \approx 1$. We use the correspondence between $\Gamma_{\mathcal{T}}^{\tilde{\mathcal{S}}_\sigma}$ and $\bar{\Gamma}_{\mathcal{T}}^{\tilde{\mathcal{S}}_\sigma}$ to define a practical $\epsilon$-proxy consistency measure as follows. We firstly re-express $\epsilon$-consistency/coherence but for $\bar{\Gamma}_{\mathcal{T}}^{\tilde{\mathcal{S}}_\sigma}$ and a different $\bar{\epsilon}$. We then trace this back to $L(\mathcal{T}, \tilde{\mathcal{S}}_\sigma)$ so a bound in terms of the consistency loss can be reported as the overall $\epsilon$-proxy. Together this yields the following:

$$\bar{\epsilon} \geq 1 - \bar{\Gamma}_{\mathcal{T}}^{\tilde{\mathcal{S}}_\sigma}$$
$$\ln \frac{1}{1 - \bar{\epsilon}} \geq -\ln(\bar{\Gamma}_{\mathcal{T}}^{\tilde{\mathcal{S}}_\sigma})$$
$$\geq L(\mathcal{T}, \tilde{\mathcal{S}}_\sigma) \tag{27}$$

and, via the relationship between $\bar{\epsilon}$ and $L(\mathcal{T}, \tilde{\mathcal{S}}_\sigma)$, we can use the consistency loss $L(\mathcal{T}, \tilde{\mathcal{S}}_\sigma)$ as a proxy measure for $\epsilon$-consistency/coherence.

## H   Spearman's Rank Correlation Analysis

A Spearman rank correlation analysis was performed between each consistency loss (Con-A, Con-I-T, Con-I-A, Con-I-R) and PRT performance. Coefficients are reported for each combination of

consistency loss and region of latent space (data-embedding, interpolation and extrapolation). Note that coefficients are not separated between $\beta$ and relation-decoder choice. Overall, each coefficients aims to characterize the PRT performance change when a relation-decoder is more or less consistent with a given partial theory, over a particular region of latent space. The key findings are reported in the main text and we tabulate the values in Table 4.

Unlike the popular Pearson correlation, the Spearman rank correlation can describe monotonic curvilinear relationships between variables. A Spearman rank coefficient varies between $-1$ and $+1$, where a coefficient $\pm 1$ indicate a perfect rank correlation. If the coefficient is negative (positive) this means a reduction (increase) in one variable corresponds with an increase in the other.