# OpenReview forum: "Formalizing Consistency and Coherence of Representation Learning"
_NeurIPS.cc/2022/Conference — NeurIPS 2022 Accept_

### Official Review · Reviewer_pBe5 · 2022-07-11

**Rating:** 4
**Confidence:** 4
**Soundness:** 2 fair
**Presentation:** 4 excellent
**Contribution:** 2 fair

**Summary:**

The paper formalizes coherence and consistency measures in ANN models that consist of an encoder ($\psi$) and relation ($\phi$) networks. To demonstrate the usefulness of these measures, they train autoencoders to additionally be coherent and consistent across two domains with respect to a single relation that was learned in only the source domain. They show that more consistent models, which implement a proposed DC model as a relation model $\phi$, are also better in zero-shot transfer to unlearned relations in the target domain. Thus, the paper demonstrates that coherence and consistency, as they formalize it, are useful for evaluating relation models.

**Questions:**

Request: the supplementary materials, and relating the work to Relation Networks.

**Limitations:**

The paper addresses its limitations well.

**Strengths And Weaknesses:**

Strengths:

1) Clear writing.

2) The DC relation-decoder seems promising for other applications.

3) The formal structure that was used to define coherence and consistency is interesting because it formally describes fundamental concepts.

4) An interesting main result, of consistency leading to better zero-shot transfer to other relations.


Weaknesses:
1) Missing supplementary materials don’t allow for a full paper evaluation.

2) The loss of eq. (9) is very similar to the established Relation Network (RN) loss [1, 2]. The paper ignores this similarity. Recognizing this similarity would have enabled comparison with results that were obtained using RN and its derivatives.

3) Following the previous point, I think that the formalism presented in the paper might not be justified. The paper could have been rewritten using only common machine learning terms such as the RN loss values and generalization errors.



[1] - Sung, F., Yang, Y., Zhang, L., Xiang, T., Torr, P. H., & Hospedales, T. M. (2018). Learning to compare: Relation network for few-shot learning. In Proceedings of the IEEE conference on computer vision and pattern recognition (pp. 1199-1208).

‏
[2] - https://en.wikipedia.org/wiki/Relation_network

---

> ### Author Response · Authors · 2022-08-02
> **Response to Reviewer pBe5**
>
> Dear Reviewer pBe5,
>
> We firstly apologise for the omitted supplementary. We provided a revision as soon as we could, which includes the appendices.
>
> We have endeavored to answer your comments about the similarities of our formalism with the Relation Networks and the justification of our formalism in the reviewer-wide response.
> Additionally, in our direct response to Reviewer jjf1 we showed how our formalism applies to standard neural networks, and in our response to Reviewer kLPG we provided an example of our formalism applied to a more complex and important learning task. We hope that these responses have addressed your concerns and we welcome further discussion.
>
> Thank you,
>
> Authors

---

> > ### Comment · Reviewer_pBe5 · 2022-08-05
> > **The manuscript still requires a revision**
> >
> > Dear authors,
> >
> > After reading other reviewers and the authors' replies, I decided to keep the rating at 4. The work is interesting but still has some flaws. Mainly, the work still does not address its relation to the literature in the manuscript . For example, I am a former user of the RN model. My main question reading the manuscript was: can I use RN to measure consistency? Would I get similar results? A comparison is needed. Regarding the formal structure that the paper presents for defining consistency and coherence, it seems that it does not provide any additional value. It provides an alternative terminology. But this alternative terminology results in things that we can describe in simpler terms. Then, what is the benefit of using it?
> >
> > The authors can amend these flaws in a revised version of the paper. I wish you all the best.

---

> > > ### Author Response · Authors · 2022-08-05
> > > **Relation Network versions/differences and how consistency can be applied to the RN from the latest paper**
> > >
> > > Dear Reviewer pBe5,
> > >
> > > Firstly, thank you for the time you have taken to review our paper. Regardless of the decision it is appreciated.
> > >
> > > After looking through the shared papers and their origins it seems there are some differences:
> > > - [1] abstract refers to "Our method, called the Relation Network (RN)". Makes a reference to similarities with [3] which also use an already termed "Relation Network" but this one is from [4].
> > > - [2] Uses Relation Networks and references [4] as the origin.
> > > - [3] Uses Relation Networks and references [4] as the origin.
> > > - [4] "introduces" Relation Networks.
> > >
> > > There may be a conflict between [1] and [4] as both introduce a Relation Network. The commonality between the RNs in [1] and [2,3,4] is the use of a relational inductive bias in their architectures, where a shared function is applied to each object-embedding pair. However [1] decodes a single value (as a relation-decoding) between each pair, but [2,3,4] produce a vector embedding that seems to capture, in some form, the "ways in which two objects are related" [2]. The relevant set of object pair vector embeddings are then aggregated and passed through a decoder, $f_{\phi}$, that provides the task-specific decoding and is also used in [3] to decode the PGM structure as a "meta-target prediction".
> > > An observation is that it is important that in [3] they found that using the meta-target prediction led to a 13.9% neutral regime test accuracy increase (and increases in other regimes). By including the PGM structure as an auxiliary, we expect that the consistency will have improved, but to test this a more explicit relation-decoding will be needed. The WReN has room to improve and it is possible that further strengthening the consistency via more explicit relation-decoding and consistency regularization would lead to further improvements. This is supported by the improved WReN performance when using disentangled panel representations [5], as our work showed that disentangled representations can improve consistency of neural network relation-decoders.
> > >
> > > We illustrated how our formalism generalises [1] in our previous response. Doing so for the RN model used in [3] will require more work as the relational representation is much less explicit. Consistency can be evaluated by unpacking the PGM structure and checking that $g_{\theta}$ and $f_{\phi}$ are together consistent with the rules that underpin these PGM structures. This would likely require explicitly decoding, from the $g_{\theta}$ output representations, the relations that these rules comprise.
> > >
> > > [1] Sung, F., Yang, Y., Zhang, L., Xiang, T., Torr, P. H., & Hospedales, T. M. (2018). Learning to compare: Relation network for few-shot learning. In Proceedings of the IEEE conference on computer vision and pattern recognition (pp. 1199-1208)
> > >
> > > [2] A. Santoro, D. Raposo, D. G. Barrett, M. Malinowski, R. Pascanu, P. Battaglia, and T. Lillicrap. A simple neural network module for relational reasoning. In NIPS, 2017
> > >
> > > [3] Barrett, D., Hill, F., Santoro, A., Morcos, A., & Lillicrap, T. (2018, July). Measuring abstract reasoning in neural networks. In International conference on machine learning (pp. 511-520). PMLR.
> > >
> > > [4] Raposo, D., Santoro, A., Barrett, D., Pascanu, R., Lillicrap, T. and Battaglia, P., 2017. Discovering objects and their relations from entangled scene representations. arXiv preprint arXiv:1702.05068.
> > >
> > > [5] Steenbrugge, X., Leroux, S., Verbelen, T. and Dhoedt, B., Improving generalization for abstract reasoning tasks using disentangled feature representations. Neural Information Processing Systems (NeurIPS) Workshop on Relational Representation Learning, Montréal, Canada., 2018.

---

> > > ### Author Response · Authors · 2022-08-09
> > > **It is not possible to measure RN’s consistency without our proposed approach**
> > >
> > > Dear Reviewer pBe5,
> > >
> > > For the sake of clarity we address your two questions: 1) Can I use RN to measure consistency? 2) Would I get similar results?
> > >
> > > In response to 1) No, RNs (as per papers [2,3,4] from previous responses) cannot be used to measure consistency without architectural modifications. RNs have no explicit mechanism to express constraints or properties among primary relations.
> > >
> > > In response to 2) An adaptation of the RN architecture would be required to evaluate consistency among learned primary relations. But we should expect to find that better consistency correlates with better results, as our experiments indicate in the case of standard NNs.
> > >
> > > We believe this answers the key questions that you raised in your review and ask you to reconsider your rating for our paper.
> > >
> > > Thank you,
> > >
> > > Authors

---

### Official Review · Reviewer_kLPG · 2022-07-12

**Rating:** 5
**Confidence:** 5
**Soundness:** 3 good
**Presentation:** 3 good
**Contribution:** 2 fair

**Summary:**

This paper addresses the problem of how logical predicates (e.g. isGreater, isSuccessor, etc.) can be learned from real-world data in a manner that transfers from one domain (e.g. images of handwritten digits) to another (e.g. images of stacked blocks). Focusing on ordinal relations, the paper first proposes a formal theory that relates first-order logic to 'soft structures' which can be learned directly from real data. This theory is then leveraged to derive formal definitions of 'consistency' (consistency of learned logical concepts within a domain) and 'coherence' (consistency of learned concepts between domains). Finally, a neural network architecture and training procedure are proposed. A particular variant of the architecture, with a novel comparator function, performs well on the transfer evaluation, and performance on this transfer evaluation is generally shown to align with the proposed definitions of consistency and coherence.

**Questions:**

My questions are outlined in the 'strengths & weaknesses' section above.

**Limitations:**

I do not envision any potential negative societal impact from this work.

**Strengths And Weaknesses:**

I have reviewed this paper before, and after reading through the new manuscript in detail, it appears that the authors have not made any changes to address the concerns that I raised in my previous review. Therefore, I am including that review below. I look forward to hearing the author's responses:

I am not familiar with any other work that attempts to build a rigorous theory connecting first-order logic with concepts learned from real-world data, so this aspect of the work was highly novel. The proposed neural architecture is well-motivated on the basis of the preceding theoretical considerations, but in the end is not particularly novel -- it amounts to an autoencoder paired with supervised training of relational concepts. The proposed comparator function is novel, but I worry that it is over-engineered for ordinal relations studied in this work, and the generic neural network relation decoder performs surprisingly well by comparison.

Overall, the novelty and significance are primarily in terms of the theoretical foundation sketched in sections 2-4. I found these sections deeply stimulating, but in the end I have significant worries about the practical relevance of the overall paradigm. The general worry is that the approach seems well-suited to the ordinal concepts studied in this work, where the formal theory governing these concepts can be neatly defined. It's unclear how the proposed approach will scale to more complex, fuzzy, real-world concepts (such as those that figure in natural language), where the specification of such a formal theory is likely impossible.

The authors may argue that their approach doesn't actually require a formal theory to be specified for the concepts in question. Strictly speaking, all that is required for training the proposed neural network model is to have training examples corresponding to positive and negative examples of these concepts that can be used for supervised learning. This doesn't necessarily require a formal theory of how these concepts relate, and therefore is perhaps more easily scalable to more complex concepts whose meaning can't be neatly defined. But this then leads me to wonder, what is the purpose of connecting this approach to a theory in terms of first-order logic? Beyond being an interesting intellectual exercise (which it certainly is), does this suggest any particular directions for learning real-world concepts in a manner that transfers better between domains? On its face, the proposed neural network architecture and training procedure constitute a rather straightforward way of learning concepts through supervised learning, and it's not clear that their development depends in any way on the accompanying theory.

Another concern is that it's not even clear to me whether concepts (in the sense of logical predicates) are needed at all, and it isn't clear what we'll do with these concepts once we have a neural system that can obtain them from real-world data. It would be helpful if the authors could sketch out how they envision this approach fitting into the broader program of AI research, and in particular how these learned concepts can be useful for downstream functionality, e.g. applied problems in computer vision or NLP.

---

> ### Author Response · Authors · 2022-08-02
> **[1/2] Response to Reviewer kLPG**
>
> Dear Reviewer kLPG,
>
> We are grateful for your continued time and input. We endeavor to answer each of your main comments in the following:
>
> **[C1]** *“I have reviewed this paper before, and after reading through the new manuscript in detail, it appears that the authors have not made any changes to address the concerns that I raised in my previous review.”*
>
> The primary changes since the previous work that you reviewed are updates to the text to improve the clarity of the motivation and to reduce the relevance of the choice of architecture in the contributions.
>
> **[C2]** *“The proposed neural architecture is well-motivated on the basis of the preceding theoretical considerations, but in the end is not particularly novel -- it amounts to an autoencoder paired with supervised training of relational concepts”*
>
> The architecture is indeed not particularly novel and this is by design - we aimed to provide a fundamental and widely applicable formalism and evaluation, and the auto-encoder architecture was a natural choice. We would additionally like to refer you to points 1 and 4 of the reviewer-wide response and to our response to reviewer jjf1, where we provide additional discussion about the applicability of this work to other architectures.
>
> **[C3]** *“It's unclear how the proposed approach will scale to more complex, fuzzy, real-world concepts (such as those that figure in natural language), where the specification of such a formal theory is likely impossible”*
>
> The key detail is that our formalism works with fragments of theories. If we have an incomplete understanding of the rules of a given concept, we can use an incomplete set of rules of a given concept and test whether the predictions are consistent with these fragments even when a full theory is not available. This allows knowledge injection even if that knowledge is incomplete and the link to a formal first-order representation means that we can explicitly express our incomplete understanding of a concept in a formal and verifiable way.
> More complex concepts, such as those encountered in NLP, while difficult to express a complete formal theory for, are still understood abstractly to some degree. This is what we mean when we refer to them as a concept (something that carries meaning at an abstract human level). For instance, words and the parts of speech that they represent (e.g. noun, adjective, verb etc) typically follow certain rules of grammatical construction.
> It would be difficult to a) fully express those rules in their entirety, and b) even if this were possible then these rules might need to be expressed in a fuzzy or probabilistic way... but that is fine. Our metrics are soft, in the sense that violations are given a numeric penalty rather than strictly disallowed. Also, we can express parts of a theory of language and use the resulting formal description to generate consistency penalties with that fragment (as we demonstrate in our experiments in the existing paper).
> We lastly note that some useful concepts, though complex, are better understood and appear widely across real-world problems and underlie certain predictions. This includes: ordinality, set relationships (contained within, intersects etc), tree-structures/taxonomies. The associated theories can be re-used as general knowledge across problems.
>
> **[C4]** “what is the purpose of connecting this approach to a theory in terms of first-order logic?”
>
> This is answered by point 3 in the reviewer-wide response.
>
> **[C5]** *“does this suggest any particular directions for learning real-world concepts in a manner that transfers better between domains”*
>
> For transfer, consistency means that we can rely on the inter-relatedness of the relation-decoders via their consistency with the rules that constrain them, in order to make sure they transfer as a system of relations. If the system of relations are consistent, then correctly predicting one means correctly predicting the others insofar as the logic allows, which makes the parent concept easier and more robust to transfer.
> In terms of directions moving forwards, the natural follow-on is to encourage consistency of models when learning different tasks, which if successful will result in a more transferable concept.
>
> (Continued in part [2/2])

---

> > ### Author Response · Authors · 2022-08-02
> > **[2/2] Response to Reviewer kLPG -  An example of a more complex concept**
> >
> > **[C6]** *“it's not even clear to me whether concepts (in the sense of logical predicates) are needed at all, and it isn't clear what we'll do with these concepts once we have a neural system that can obtain them from real-world data. It would be helpful if the authors could sketch out how they envision this approach fitting into the broader program of AI research, and in particular how these learned concepts can be useful for downstream functionality”*
> >
> > Reviewer jjf1 proposed that we restructure the preliminaries and include an intuitive example to introduce the required notation/concepts and we believe this can be beneficial for **C6** also. In the following we present a candidate for this example:
> >
> > **Example 1. Application of the formalism to a more complex concept:**
> > Take a domain of text-based résumés. The task is to predict whether a person with a given résumé is a good hire, i.e. decoding the relation $\textsf{isGoodHire}(x)$. Suppose we do this with a neural network that first encodes a résumé and produces a binary prediction for  $\textsf{isGoodHire}(x)$. We would want to ensure that the obtained concept of hireability is unbiased. We can give rule-based examples of biases, such as:
> >
> > $\textsf{isGoodHire}(x) \rightarrow \textsf{isGenderA}(x)$  (hireability implies a gender)
> >
> > $\textsf{isGoodHire}(x) \rightarrow \textsf{isEthnicityB}(x)$ (hireability implies an ethnicity)
> >
> > This prompts us to decode these additional relations from the résumé encoding, and then we can apply our metric to evaluate whether the bias holds (i.e. if the consistency loss is small with respect to the available theory fragments).
> > Providing we have a relation-decoder that has a coherent notion of hireability with respect to unwanted biases, we can (a) predict hireability with confidence that the model is not biased (insofar as the available theory suggests), and (b) use this collection of relation-decoders to ensure any subsequent application of hireability to new data-domains (e.g. résumés that include photos) which require new data-encoders is unbiased by employing them as fixed parameter relation-decoders.
> > The natural follow-on work is to use the consistency loss to regularize for the desired outcome, in this case we would maximize the loss against these biased rules.
> >
> > We hope that our response answers your comments and we welcome further discussion.
> >
> > Thank you,
> >
> > Authors

---

> > ### Comment · Reviewer_kLPG · 2022-08-08
> > **Followup**
> >
> > Thanks very much to the authors for this thorough rebuttal. The hireability example does a good job of illustrating how this approach might be extended and made more generally useful. And I appreciate the efforts to relate the proposed approach to a wider set of architectures.
> >
> > It seems there are two distinct contributions in this work. One is the proposed architecture and DC module. As I mentioned before, these strike me as not particularly novel, and also not obviously dependent on the formalism, other than that they serve as a testbed for the proposed losses. In my read of the manuscript, it seemed this aspect (the architecture) was emphasized as a main contribution, and something that directly followed from the proposed formalism. However, in the rebuttal, it sounds like the authors view the architecture as merely a testbed for the proposed losses, which I think makes more sense. It would be good to revise the paper accordingly. The second contribution then is the proposed coherence and consistency losses themselves. It is very good to hear that, in principle, these do not require fully specified theories for any given domain, and could still work with only fragments. This is a very desirable feature as this seems certain to be the situation with most real-world problems. However, it would be much more convincing if the authors could include an example demonstrating how well this would actually work in practice, ideally in a more complex domain than ordinal relations.
> >
> > On balance, I believe the attempt to address a broader range of architectures, and the clarification of the aims of the work (i.e. the contributions of the architecture vs. losses) merits a score increase, but it is still borderline. A working example in a domain where only fragments of a formal theory are available would push the work to a clear accept, though I assume that is not feasible for this conference.

---

> > > ### Author Response · Authors · 2022-08-09
> > > **Thank you + a last note**
> > >
> > > Dear Reviewer kLPG,
> > >
> > > Thank you for reconsidering your rating. We would like to note briefly that results using different fragments within the concept of ordinality (but having broader relevance) are already included in the paper. Figure 3 shows consistency losses for the transitivity, reflexivity and asymmetry fragments over each relation, which are considered independently from the other consistency losses. There may be a lot to unpack in Figure 3 with more information provided in Appendix F showing the truth-tables for different theory fragments. We will work on the revised version of the paper to make the presentation and results as clear as possible within the available page limits.
> > >
> > > Thank you,
> > >
> > > Authors

---

### Official Review · Reviewer_jjf1 · 2022-07-16

**Rating:** 7
**Confidence:** 2
**Soundness:** 4 excellent
**Presentation:** 4 excellent
**Contribution:** 3 good

**Summary:**

This paper is motivated by the formal concepts of coherence and consistency in neuro-symbolic AI, and the fact that such concepts have not been formalized in the connectionist paradigm. It claims that these concepts are important for assessing transfer learning performance. This is demonstrated by formalization of these concepts for neural networks, proposal of the "relation-decoder" model, and transfer learning experiments to show that good transfer performance is associated with coherence being maintained across domains and can be predicted by ability to maintain consistency over unseen inputs.

**Questions:**

- In the space of transfer learning architectures, and NN architectures in general, what all falls under this formalization paradigm?

**Limitations:**

The paper is upfront about its technical limitations. Based on the last couple sentences of sec7, it is likely that the authors and I disagree on the importance of having other architectures considered in this paper, but the acknowledgment is valuable nonetheless.

The discussion of social limitations leaves something to be desired, but I commend the authors on engaging with it at all when a lot of core theoretical papers get away with a generic sentence. The mention of logic, biases, and disentanglement are interesting and merit development; the claim about trustworthiness of coherent concepts is also intriguing and would benefit from examples and some investigation.

To go deeper, there are theoretical and empirical STS-type works that would engage further with the ideas of logical validity and absolutism as well. However, that would be getting beyond the scope of this type of statement. Ultimately these are **not** knocks on the paper - I only have words for this section because the authors chose to say something (anything) more than "N/A".

**Strengths And Weaknesses:**

*Originality*: to my knowledge, this paper is original. Beyond the labels, I'm not familiar with these specific formulations being used as metrics for neural networks. This paper spends time formally defining predicate logic concepts that may not need defining (at least not in the main paper) as they are very fundamental, but the soft structure formalization is technically new to me.

*Quality*: this paper takes a well-scoped problem and lays out a clear strategy to show it. The goals seem to be to present formal definitions of consistency and coherence that are meaningful in the connectionist paradigm, create a transfer learning framework from which these metrics can be calculated, and show that they are useful indicators of transfer learning ability - presumably, that they predict standard performance metrics while also providing more insights. I think this is done successfully - the setup and experiments are very much called for by the aims.

The only area for improvement I would bring up is usefulness beyond the relation-decoder framework - if the metrics only work for not only a narrow set of architectures but specifically the architectures in this paper, then it seems the paper should be more about the architecture than the metrics (though this would require different experiment analysis and to some degree different experiments).

*Clarity*: This paper is quite well-written. Unlike most papers, the paper roadmap is clear early on in reading. I felt I had a pretty good mental model for the contributions after the intro, and could consider the rest of the paper from that perspective.

Some areas for improvement include a less dense and better-illustrated formalization section - this may be more about the paper than the research, but a lot of space is devoted to formalization steps prior to the structure and soft-structure definitions that may not be necessary for someone familiar with basic predicate logic, which is likely most of the audience. Some of the more composite concepts (signature, interpretation) would benefit from examples and intuitive explanation, since they seem to be more familiar and easy to grasp than they are presented as being.

*Significance*: This paper presents novel (to my knowledge) metrics for assessing transfer performance. As machine learning moves toward scale and ubiquity, transfer performance and robust evaluation are both important areas in my opinion, so these are posed to be valuable contributions. My primary concerns about significance are related to my comments in quality - it's unclear to me how far these metrics can apply. While other architectures may fall under the same framework mathematically, I think the burden of assessing scope is on the paper since the message is ultimately about the value of the metrics.

---

> ### Author Response · Authors · 2022-08-02
> **Response to Reviewer jjf1**
>
> Dear Reviewer jjf1,
>
> We would firstly like to thank you for your positive comments. We are glad that you appreciate the paper and from your comments it is clear that you understand the paper motivation, structure and outcomes.
>
> Your comment on the applicability of the metrics to a wider set of architectures is important. We have discussed this in point 1 of the reviewer-wide response but we would like to add more here. Firstly, we chose the auto-encoder architecture because it clearly exposes the principle parts of many connectionist systems - an encoder, an embedding layer, and a decoder. However, in terms of components, a simple neural network (NN) works analogously: for an $n$ layer NN with $n>m$ we can treat the layers up to $m$ as an encoder, the layer $m$ as the embedding and the layers thereafter as a decoder (see Figure 1 in [1]). In this way the formalism is applicable to a very wide range of architectures.
>
> Including the auto-encoder’s reconstruction is not required, but it is useful in a broader sense, because it allows us to retain input-data information when decoding for only certain relations. If we only had relation-decoders, then the irrelevant information would be lost. The relation-decoders are trained in a weakly-supervised manner and so we can view them as an auxiliary decoding next to the reconstruction backbone. We can thus apply relation-decodings and consistency losses as auxiliaries to other backbone tasks. We give an intuitive example of this comment in the response to Reviewer kLPG titled: “An example of a more complex concept” (this example also shows our reasoning regarding the trustworthiness of coherent concepts).
> Following your suggestion we would like to move the bulk of the preliminaries to the appendix and use an intuitive example like this to present the motivation and required preliminaries in a more accessible manner.
>
> Thank you,
>
> Authors
>
> [1] Shwartz-Ziv, R. and Tishby, N., 2017. Opening the black box of deep neural networks via information. arXiv preprint arXiv:1703.00810.

---

### Author Response · Authors · 2022-07-27
**Thank you for the reviews + revision submitted with supplementary**

Dear Reviewers,

Firstly, thank you for your reviews and for the time you have taken to do them. We are in the process of preparing our rebuttal and aim to have this prepared and submitted soon.

In the meantime, we have submitted a paper revision that includes the supplementary pages. Of particular importance are a proof of the consistency loss we use in the paper and a deeper explanation of the DC relation-decoder.

Sincerely,
The Authors

---

### Author Response · Authors · 2022-07-29
**To all reviewers - addressing the main points that were common among the reviewers**

We aim to provide a full rebuttal by Monday but to open the discussion we address here the main points that were common among the reviewers.


**1. On the applicability of our framework to different architectures:**
The auto-encoder architecture used in our experiments was a natural choice for the comparative evaluations that were carried out. More generally, all that is needed for our proposed consistency loss to be applied is an encoder and one or more relation-decoders. Any architecture that predicts categorical or binary outputs should be applicable as long as the initial layers act as an encoder producing an embedding that is used by the output layer to give us the relation decodings. Thus, the formalism is not restricted to the architecture used in the paper. As another example, we discuss how the formalism would apply to the Relation Network architecture in point 4.

**2. On the use of more complex concepts:**
Please note that our formalism does not require a complete logical theory to be fully specified. It works on fragments of theories (just the concepts that the user may want to check for consistency, as in the case of transitivity which we investigate in Fig. 3-bottom). Transitivity, ordinality, and counting may be considered to be relatively complex already. We can query for any concept specifications that are available to obtain consistency evaluations against those fragments of logical theory.
Certain fragments will be common across learning tasks. Humans make use of intuitive physics (conservations, symmetries) and psychology **[1]**. If we include decodings that cover these against a target learning task we can re-use the fragments and ensure the learning task is consistent.

**3. On the value of a formal language and specification in logic:**
We use first-order logic to formally express the learning task and relevant concepts with a precise semantics. Our consistency loss then provides a metric to use in the evaluation of model performance going beyond accuracy values. As shown in the paper this is important when evaluating transfer learning as the consistency loss on our evaluations anticipates the transfer learning performance to some degree. This is key because consistency loss on different embedding distributions can be evaluated before the transfer task is specified. We can reduce the formality of the presentation of the logic in the paper by moving the formal definitions to the supplementary materials and giving an informal intuition by examples of the use of logic in the main body of the paper thus hopefully making the paper more accessible to a wider audience.

(Continued in the following official comment below)

[1] Lake, B.M., Ullman, T.D., Tenenbaum, J.B. and Gershman, S.J., 2017. Building machines that learn and think like people. Behavioral and brain sciences, 40.

---

> ### Author Response · Authors · 2022-07-29
> **[Continued] To all reviewers - addressing the main points that were common among the reviewers**
>
> (Continuing from point 3 above)
>
> **4. On the differences between our work and Relation Networks (RN):**
> Our loss is based on cross-entropy because it is a common choice. RN uses a squared error loss and the RN authors remark that this is less common. RN looks at relations of the form “$g_\phi = \textsf{isSameClass}(x_i, x_j)$”, most similar to “$\textsf{isEqual}(x_i, x_j)$” in our case, but not applicable to the other relations that we consider (e.g. successor/predecessor or greater/less). The RN paper is written in terms of labels corresponding to mutually exclusive classes, whereas for us we relax this requirement to support arbitrary constraints on predictions which our experiments indicate to be an important distinction.
>
> The best way to consider their differences is to evaluate how we can match learning setups. To reconcile the two learning tasks choose a single domain $\mathcal{X}$ of numbers but divide it into two sets $\mathcal{X_i}$ and $\mathcal{X_j}$ where $i$ contains digits {3, 4, 5, 6} and $j$ contains {0, 1, 2, 7, 8, 9}. Then extract the support and test sets from $\mathcal{X_j}$ such that they are disjoint class instances. To reconcile the difference in models, we show that this is just a change of implementation but the underlying components are analogous: we swap $f_{\psi}$ for $\psi_{\mathcal{X}}$ (same domain, so we use the same encoder model) and $g_{\phi}$ to $\phi$ and we arrive at the same model, finally omit the use of a reconstruction decoder. For relation decoders in the RN domain we only need $\phi =$ {$ \phi_{isEqual} $}. For the encoder $\psi_\mathcal{X}$ we can use the same encoding pipeline and model as the relation network (i.e. $f_\psi$ or $f_{\psi_1}$ and $f_{\psi_2}$ together).
>
> We have shown simply that our framework generalises the RN. But, moreover, our metric is useful in the RN setting as well: we can use a mutual exclusivity constraint across the classifications, i.e. if $x_i$ is equal to class $c$ it cannot be equal to any others. Trivially, a model that performs well on the RN one-shot learning task would need to obtain good consistency loss on mutual exclusivity across classes in the target embeddings. We note that we can also apply common consistency-individual (Con-I) theory fragments to evaluate whether the isEqual relation exhibits the correct characteristics in RN (if this is not the case then we know that there are error-cases i.e. $\textsf{isEqual}(x,y) \land \neg \textsf{isEqual}(y, x)$ means at least one inference is wrong).
>
> ___
>
> We hope that the above addresses the main points that were common among the reviewers and we welcome further discussion.

---

> > ### Comment · Reviewer_pBe5 · 2022-08-04
> > **Relation Networks are not constrained to isSameClass relations**
> >
> > RNs are not restricted to isSameClass relations. They were shown to successfully find rules such as greaterThan and even more complicated rules, such as boolean-logic rules (AND, OR) [1].
> >
> > [1] - Barrett, D., Hill, F., Santoro, A., Morcos, A., & Lillicrap, T. (2018, July). Measuring abstract reasoning in neural networks. In International conference on machine learning (pp. 511-520). PMLR.‏

---

> > > ### Author Response · Authors · 2022-08-05
> > > **Response to PBe5 "Relation Networks are not constrained to isSameClass relations"**
> > >
> > > In the first paper only the relation “isSameClass” was explicitly decoded. The newly shared paper is based on a different "Relation Network" (with a different origin), which models relations in a significantly different way to the first paper you shared. This different architecture is applied to a complex task that involves these more complicated relations.
> > >
> > > The similarity between shared papers is that they define a task and then seek to measure the reasoning on that task. Our paper proposes a principled approach that goes the other way round: define the reasoning metric first so it can be applied to any task/dataset.
> > > It is well-known that standard NNs can also "successfully find rules" including "complicated rules, such as Boolean-logic rules (AND, OR)". The fact that RNs (and many other neuro-symbolic approaches) also do this makes our proposed metric all the more justified.

---

### Author Response · Authors · 2022-08-05
**Response to Reviewer pBe5's decision**

Dear Reviewers,

Reviewer pBe5 seems to have rejected our paper because we do not compare with the Relation Network (RN). So far we have been presented two RN versions with significantly different architectures - in our rebuttal we have endeavoured to illustrate how our formalism applies to both. In our paper, we do compare with standard neural networks and various other models. There are many other neuro-symbolic approaches, precursors of RN, which encode relations. We cannot compare with them all but we have illustrated how our formalism generalises across many architectures.

All in all, we have tried to argue that, rather than direct comparisons on hand-picked tasks, datasets and architectures, what matters here is the formalization and general applicability of the principles of consistency and coherence.

Thank you,

Authors

---

### Meta-Review · Area_Chair_fsBQ · 2022-08-26

**Recommendation:** Accept
**Confidence:** Less certain

**Metareview:**

This paper has slightly mixed reviews, trending toward a weak accept overall. The paper's topic and approach are overall novel, supported by good experiments. It's an interesting paper that will likely spark further investigation and inspire other research. There is still some disagreement among the reviewers, with recommendations ranging from accept to weak reject. In particular, the reviewers believe that, while overall well written, the paper could still use some improvements in presenting the relationship among the ideas and the architecture. In particular, the authors are encouraged to revise their presentation to better present "the architecture as merely a testbed for the proposed losses", as suggested in the discussion below. These revisions are easily made. The reviews also suggested improvements in a working example, which the authors' stated already exists in Fig 3 and admit is rather dense. It would benefit the paper to include a more detailed and clear work-through of the example, possibly in summary in the main paper and then in detail in the appendix.

Side note: the authors' privately expressed concerns were taken into consideration by the meta-reviewer in evaluating this paper.

**Award:**

No

---

### Decision · Program_Chairs · 2022-09-14

Accept